# Robust uncertainty estimates with out-of-distribution pseudo-inputs training

## Abstract

Probabilistic models often use neural networks to control their predictive uncertainty. However, when making *out-of-distribution (OOD)* predictions, the often-uncontrollable extrapolation properties of neural networks yield poor uncertainty predictions. Such models then don't *know what they don't know*, which directly limits their robustness w.r.t unexpected inputs. To counter this, we propose to explicitly train the uncertainty predictor where we are not given data to make it reliable. As one cannot train without data, we provide mechanisms for generating *pseudo-inputs* in informative low-density regions of the input space, and show how to leverage these in a practical Bayesian framework that casts a prior distribution over the model uncertainty. With a holistic evaluation, we demonstrate that this yields robust and interpretable predictions of uncertainty while retaining state-of-the-art performance on diverse tasks such as regression and generative modelling.

## 1 Introduction

Neural networks generally extrapolate arbitrarily (Xu et al., 2020), and high quality predictions are limited to regions of the input space where the networks have been trained. This is to be expected and is only problematic if the associated predictions are not accompanied with a well-calibrated measure of uncertainty. If a neural network is used for estimating such a measure of uncertainty, we, however, quickly run into trouble, as the reported uncertainty then exhibits arbitrary behaviour in regions with no training data. Alarmingly, these are exactly the regions where evaluating the uncertainty is most important to the safe deployment of machine learning models in real world applications (Amodei et al., 2016). One potential solution is to avoid using directly the output of neural networks for predicting uncertainty, and let it emerge from another mechanism, e.g. an *ensemble* (Hansen & Salamon, 1990; Lakshminarayanan et al., 2017) or some notion of *Monte Carlo* (MacKay, 1992; Gal & Ghahramani, 2016). Here we explore the alternative view that the networks should simply be trained where there is no data. But can we train without data? The Bayesian formalism often does so implicitly: most *conjugate priors* can be seen as additional training data (Bishop, 2006), e.g. in Gaussian models, a mean prior $\mathcal{N}(\mu_0, \sigma_0^2)$ can be realised by additional training data of $\mu_0$ with $\sigma_0^2$ setting the amount of observations. Placing a prior over the output of a neural network can, thus, be interpreted as additional training data. Unfortunately, this view is not practical as it implies

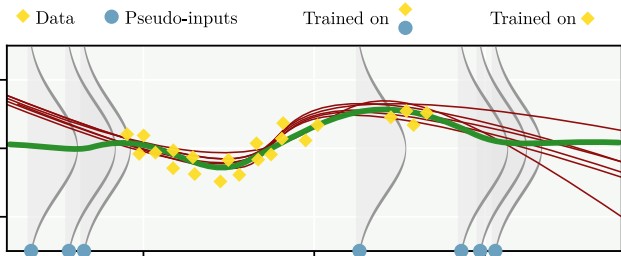

Figure 1: Pseudo-inputs are generated out of distribution, and there we train towards a prior (grey density).

additional data *for all* possible inputs to a neural network, resulting in infinite data. Our approach is simple: we locate regions of low data density in *input space* and implicitly place observations here in *output space* by minimising an appropriate KL divergence towards a prior (see Fig. 1). The result is a simple algorithm that significantly improves uncertainty estimates in both regression and generative modeling.

## 1.1 Background and related work

The predictive performance of machine learning models has drastically increased in the past decade, but the quality of the accompanying uncertainties have not followed. Uncertainties are reported as being miscalibrated (Guo et al., 2017) and overconfident (Lakshminarayanan et al., 2017; Hendrycks & Gimpel, 2016). Some models even see higher likelihoods of out-of-distribution than in-distribution data (Nalisnick et al., 2019; Nguyen et al., 2015; Louizos & Welling, 2017).

**Neural networks** commonly output distributions which gives a notion of predictive uncertainty. Classifiers trained with *soft-max* is an ever-present example of such. These predictions are generally observed to be *overconfident* (Lakshminarayanan et al., 2017; Hendrycks & Gimpel, 2016) and to carry little meaning outside the support of the training data (Skafte et al., 2019; Lee et al., 2017). The latter is an artifact of the hard-to-control extrapolation that comes with neural networks (Xu et al., 2021). In general, since extrapolation is difficult to control, uncertainties predicted by neural networks will exhibit seemingly arbitrary behavior outside the support of the data, yielding untrustworthy results.

**Mean-variance networks** for regression (Nix & Weigend, 1994) model the conditional target density as a Gaussian $p(y|x) = \mathcal{N}\left(y|\mu(x), \sigma^2(x)\right)$ with mean and variance predicted by neural networks. The predictive uncertainty is generally accurate in regions near training data, but otherwise unreliable (Hauberg, 2019). To counter this, Arvanitidis et al. (2017) and Skafte et al. (2019) proposed variance network architectures to enforce a specified extrapolation value, but these heuristics tend to be difficult to tune, and lack principle. Mean-variance networks have seen a recent uptake within generative modeling, where they are used as an *encoder* distribution in *variational autoencoders (VAEs)* (Kingma & Welling, 2013; Rezende et al., 2014).

**Which uncertainty?** A commonly called-upon dichotomy (Der Kiureghian & Ditlevsen, 2009) is that the uncertainty of a model's *prediction* can be decomposed into the uncertainty of the *model (epistemic)* and of the *data (aleatoric)*. The epistemic uncertainty can be lowered by increasing the amount of data, simplifying the model or otherwise reducing the complexity of the learning problem. The aleatoric uncertainty, on the other hand, is a property of the world, and cannot be changed; no prediction should ever be more certain than the uncertainty displayed by the associated data.

**Bayesian methods** are often used to quantify uncertainty due to their explicit formulation of uncertainty. *Gaussian processes (GPs)* (Rasmussen & Williams, 2005) provide an elegant framework that provide state-of-the-art uncertainty estimates, but, alas, the corresponding mean predictions are often not up to the standards of neural networks. GPs are tightly linked to *Bayesian neural networks (BNNs)* (Neal, 2012) that place a prior over the network weights and seek the corresponding posterior. Despite advances in *variational* and *Laplace approximations* (Graves, 2011; Kingma & Welling, 2013; Blundell et al., 2015; Daxberger et al., 2021), *expectation propagation* (Hernández-Lobato & Adams, 2015; Hasenclever et al., 2017), or *Monte Carlo* methods (Welling & Teh, 2011; Springenberg et al., 2016), training BNNs remains difficult. Furthermore, the predictive uncertainty seems dependent on the degree of approximation and is thus controlled by the available compute power.

**Ensemble methods** have long been used to produce aggregated predictions with uncertainty estimates (Hansen & Salamon, 1990; Breiman, 1996). *Deep ensembles* (Lakshminarayanan et al., 2017), a collection of differently initialized networks trained on the same data, are generally reported as state-of-the-art for uncertainty quantification in deep models (Thagaard et al., 2020; Ovadia et al., 2019). As the models in the ensemble are trained on overlapping data, they are correlated, which influence the ensemble uncertainty in ways that remains unclear (Breiman, 2001). *Monte-Carlo dropout* (Gal & Ghahramani, 2016) casts dropout training (Srivastava et al., 2014) as an ensemble model. It is computationally cheap, but experiments (Ovadia et al., 2019; Skafte et al., 2019) show that the increased correlation of ensemble elements causes overconfidence.

**Robustness to distribution shift** is part of a well-behaved uncertainty predictor (Ovadia et al., 2019) and must be evaluated accordingly. For out-of-distribution detection, Liang et al. (2017) proposes a

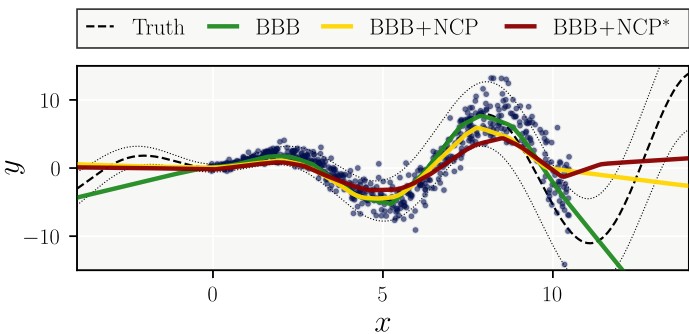

Figure 2: Noise contrastive priors improve uncertainty estimates, but overregularize the mean. BBB+NCP*
indicates Bayes by backprop with greater NCP regularisation

pre-processing perturbation step inspired by adversarial attacks (Goodfellow et al., 2014b) that helps
distinguish in-distribution and out-of-distribution inputs. Hendrycks et al. (2018) used a *Generative
Adversarial Network (GAN)* (Goodfellow et al., 2014a) to generate out-of-distribution pseudo-inputs that
appear in a regularizing term in the loss function, called *outlier exposure*, to enhance the predictor's ability
to discriminate out-of-distribution inputs (Lee et al., 2017; Dai et al., 2017).

**Out-of-distribution pseudo-inputs** can improve uncertainty estimates. Deep ensembles trained to
maximise diversity on such pseudo-inputs can display high uncertainty outside of their training support
(Jain et al., 2020), under the strong assumption that uniformly distributed pseudo inputs can accurately
capture the out-of-distribution support. High entropy priors on predictions, $p_{\text{prior}}(y|x)$, conditioned on
OOD pseudo-inputs sampled from $p_{\text{prior}}(x)$, have also successfully regularised the uncertainty predictions
of Bayesian models (Hafner et al., 2018; Malinin & Gales, 2018). But these negatively affect the *mean*
predictions compared to contemporary regularisation techniques (Srivastava et al., 2014; Ioffe & Szegedy,
2015).

## 1.2 Robust uncertainty estimates

**Notation.** Let the observed variable $x \in \mathcal{X}$ follow the data generating distribution $p_{\text{data}}(x)$, only known
through the training dataset of $N$ i.i.d samples $\mathcal{D}_{\text{train}} = \{x_n\}_{n=1}^N$. In the case of supervised learning, the
observed variables $x = (x, y)$, with $x \in \mathbb{R}^d$ being the input and $y \in \mathbb{R}^{d'}$ the target for the model, follow the
joint decomposition $p_{\text{data}}(x, y) = p_{\text{data}}(y|x)p_{\text{data}}(x)$. The proposed probabilistic model $p_\theta(x)$, whose weights
are indicated by $\theta$, aims to accurately emulate $p_{\text{data}}(x)$.

**Practical problems in variance estimation.** Gaussian likelihoods in the form of $p_\theta(x) = \mathcal{N}\left(x|\mu_\theta(x), \sigma_\theta^2(x)\right)$ are widely adopted to model continuous covariates. Real world data cannot be
expected to be *homoscedastic*, i.e constant throughout input space, and thus the predictive uncertainty, $\sigma_\theta(x)$,
most often uses neural networks to map continuously the observed $x$ onto the parameter space. Beyond
the well-known unreliable extrapolation properties of neural networks, this parametrisation of predictive
uncertainty is hamstrung by serious defects. Firstly, the predictive variance scales the learning rates of
the mean and variance updates by $1/2\sigma_\theta^2(x)$, resulting in a bias for data regions with low uncertainty (Nix &
Weigend, 1994). Secondly, the maximisation of the modelled likelihood is particularly sensitive to scarce data,
as local gradient updates for the variance point towards the then undefined *maximum likelihood estimate
(MLE)* (Skafte et al., 2019). Lastly, such model's likelihood is ill-defined (Mattei & Frellsen, 2018b), as it
can without bound increase when the variance estimates collapse towards a detrimental 0. Overall, the naive
maximisation of model likelihood is insufficient to generate robust and well-behaved uncertainty estimates,
and practical implementations rely on an arbitrary lower threshold of the predictive variance.

**Ensembles.** The arbitrariness of the extrapolation of single mean-variance networks precludes any guarantees
of robust uncertainty estimates. Previous contributions thus adopt a mixture of Gaussians as their predictive
density. Whether discrete (Lakshminarayanan et al., 2017; Gal & Ghahramani, 2016; Jain et al., 2020) or

continuous (Hafner et al., 2018), they commonly result in variance predictions expressed as a function of the mean. As shown in Fig. 2, this dependency creates a trade-off that limits the ability to improve the robustness of the uncertainty predictions without sacrificing some of the model's mean predictive power.

**Student-t likelihood.** Skafte et al. (2019) notably adopts a Gamma distributed precision, $\lambda \sim \Gamma(\alpha, \beta)$, as the conjugate of an unknown precision for a Gaussian, to yield a non-standard Student-t distributed marginal likelihood[1]. This infinite mixture of Gaussians is known to offer a more robust likelihood, especially in the scarce data regime (Gelman et al., 2013),

$$p_\theta(\mathrm{x}) = T\left(\mathrm{x}|\nu = 2\alpha, \hat{\mu} = \mu, \hat{\sigma} = \sqrt{\beta/\alpha}\right). \tag{1}$$

Its variance $\mathrm{Var}[\mathrm{x}] = (\beta/\alpha)\cdot(\alpha/(\alpha-1))$ is explicitly decomposed into an aleatoric $\beta/\alpha$ and an epistemic term[1] $\alpha/(\alpha-1)$ (Jørgensen, 2020, p16), and offers a direct verification of whether a model knows what it knows.

**Variational variance *(VV)*.** Stirn & Knowles (2020) assumes a latent model precision $\lambda$. It is generated by a prior $p(\lambda)$ and its posterior is approximated variationally by the family of Gamma distributions, conditioned on the inputs to reflect heteroscedasticity. Through *amortized variational inference (AVI)* (Kingma & Welling, 2013) neural networks $f_\phi$ map to the posterior parameters from data, $q(\mathrm{z}|f_\phi(\mathrm{x}))$. As such, variational variance preserves the modelling capacity and robustness of the Student-t marginal likelihood, without modifying its parameter architecture, while the definition of a prior over the precision induces a more robust training objective. Assuming the precision is the unique latent code, the *evidence lower bound (ELBO)*,

$$\mathcal{L}(q; \mathrm{x}) = \mathbb{E}_{q(\lambda)}\left[\log p(\mathrm{x}|\lambda)\right] - D_{\mathrm{KL}}\left(q(\lambda|\mathrm{x}) \,\|\, p(\lambda)\right) \tag{2}$$

$$= \frac{1}{2}\left(\psi(\alpha) - \log\beta - \log(2\pi) - \frac{\alpha}{\beta}(\mathrm{x} - \mu)^2\right)$$
$$- D_{\mathrm{KL}}\left(q(\lambda|\mathrm{x}) \,\|\, p(\lambda)\right), \tag{3}$$

takes the form of a regularised log-likelihood. It penalises predicted variances that would unrealistically get arbitrarily close to either the detrimental limits of 0 or $\infty$, reducing the concerns regarding the ill-definition of the objective. Additionally, the scaling effect of the learning rates of the likelihood parameters is reduced. Naturally, the effect of the regularisation will be highly dependent on the prior selected. Here, because we are mostly interested in enforcing a constant desired uncertainty extrapolation, we adopt an homoscedastic Gamma distributed prior, $p(\lambda) = \Gamma(\lambda|a, b)$, that matches the level of uncertainty observed in data.

## 2 Out-of-distribution pseudo-inputs

### 2.1 Dissipative loss

In variational variance, due to AVI, the uncertainty is controlled by $\alpha$ and $\beta$, the independent parameter maps of the posterior distribution, $\mathrm{Var}[\mathrm{x}] = \beta(\mathrm{x})/(\alpha(\mathrm{x}) - 1)$. The unreliable extrapolation properties of NNs therefore directly challenge the robustness of the method's uncertainty estimates outside of its training support.

Inspired by outlier exposure (Hendrycks et al., 2018) and noise contrastive priors (Hafner et al., 2018), we include deliberately generated out-of-distribution pseudo-inputs, $\{\hat{\mathrm{x}}_k\}_{k=1}^K$ where $\hat{\mathrm{x}}_k \sim p_{\mathrm{out}}(\mathrm{x})$, in the training of our variational objective to constrain the extrapolation of the posterior parametrisation. The optimal variational objective $q^*$ is chosen such that it minimises our proposed *dissipative loss* over the combined dataset $\mathcal{D} = \mathcal{D}_{\mathrm{train}} \cup \mathcal{D}_{\mathrm{out}}$, where $\mathcal{D}_{\mathrm{out}} = \{\hat{\mathrm{x}}_k\}_{k=1}^K$,

$$\mathrm{L}(q; \mathcal{D}) = -\left[\mathcal{L}_{\mathrm{in}}(q; \mathcal{D}_{\mathrm{train}}) + \mathcal{L}_{\mathrm{out}}(q; \mathcal{D}_{\mathrm{out}})\right]. \tag{4}$$

The in-distribution component of the loss function $\mathcal{L}_{\mathrm{in}}(q; \mathcal{D})$ naturally arises as the standard ELBO over the training set. The out-of-distribution component $\mathcal{L}_{\mathrm{out}}(q; \mathcal{D})$ operates on a fundamentally different source of data. As the only information available for the pseudo-inputs is that they are OOD, we assert for them a

---

[1]See Sec. I. of the supplementary materials.

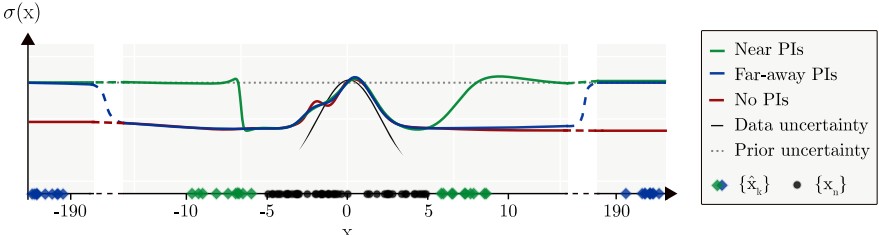
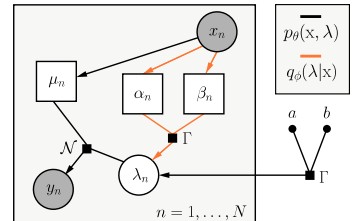

Figure 3: Predictive uncertainty of VV for different pseudo-inputs distributions. Training data is generated uniformly with $\mathrm{Var}[\mathrm{x}] \propto \exp\left(-0.5(||\mathrm{x}||/s)^2\right)$.

Figure 4: Graphical model for variational variance regression.

constant, non-informative likelihood $p(\hat{x}|\lambda) = c$, that has thus no influence on optimisation. This is similar to *censoring* (Lee & Wang, 2003) where different likelihoods are used for observations with different properties. This simplifies the pseudo-inputs generation process, which no longer depends on modelling a likelihood, as is done by e.g Hafner et al. (2018), where the prior on targets is chosen as Gaussian data augmentation. As a result, the dissipative loss becomes,

$$
\begin{aligned}
\mathrm{L}(q; \mathcal{D}) = & -\sum_{\mathrm{x}\in\mathcal{D}_{\mathrm{train}}} \mathbb{E}_{q(\lambda|\mathrm{x})}\left[p_\theta(\mathrm{x}|\lambda)\right] - D_{\mathrm{KL}}(q(\lambda|\mathrm{x}) \,||\, p(\lambda)) \\
& + \sum_{\hat{\mathrm{x}}\in\mathcal{D}_{\mathrm{out}}} D_{\mathrm{KL}}(q(\lambda|\hat{\mathrm{x}}) \,||\, p(\lambda)).
\end{aligned}
\tag{5}
$$

It shares the same motivating intuition as the *confidence loss* of Lee et al. (2017), which pushes a soft-max classifier towards the uniform distribution on OOD pseudo-inputs, and completes variational variance with a principled mechanism to learn robust variance estimates with the desired extrapolation properties. The predictor is indeed forced to match our high-entropy uncertainty prior expectations on out-of-distribution samples while learning the low-entropy covariate dependent distribution, hence the name of dissipative. The reliance of the model's predictive uncertainty on its mean predictions implies that it is primordial here to safeguard its generative performance. Previous contributions adopting a similar dual loss function (Lee et al., 2017; Jain et al., 2020; Hafner et al., 2018) contaminate the uncertainty regularisation with mean predictions, forcing joint learning of both loss terms and jeopardising the model mean predictive power. Conversely, the dissipative loss guarantees its conservation with the implementation of a split training procedure (Skafte et al., 2019); its modularity allows the application of the out-of-distribution regularisation only after the model's mean has been trained, which we view as a key conceptual advantage of our proposal.

## 2.2 Pseudo-input generators (PIGs)

Minimising the posterior KL divergence OOD requires an efficient sampling procedure of pseudo-inputs. As exposed in Fig. 3, their generation should leverage a-priori knowledge about $p_{\mathrm{data}}(\mathrm{x})$ to resolve the undefined nature of $p_{\mathrm{out}}(\mathrm{x})$. In this simple regression case, we show the predictive uncertainty of variational variance models trained on artificial heteroscedastic data. We use a prior uncertainty level that matches the maximum of the data uncertainty. As anticipated, without pseudo-inputs, the model extrapolates uncertainty to a constant, arbitrary level, and only the introduction of pseudo-inputs near the training data results in the desired uncertainty extrapolation. Reassuringly, this suggests that we do not need to regularise our model's extrapolation in the entire out-of-distribution space, as suggested by Jain et al. (2020). Instead, we can focus on the simpler task of generating pseudo-inputs in low-density regions of the input space that neighbours training data, as they can enforce correct extrapolation in the rest of the out-of-distribution space. Lee et al. (2017) gives supporting arguments for classification, and empirical results shows that this intuition generalises to higher dimension experiments[2].

Recent contributions have relied on GANs for generating a useful representation of $p_{\mathrm{out}}(\mathrm{x})$ (Lee et al., 2017; Dai et al., 2017). Although conceptually intuitive, GANs incur a heavy computational burden and

---

[2]Further experiments are in appendix II..1.

induce serious practical challenges as a result of the instability of their training (Shrivastava et al., 2017). Furthermore, as one need to understand what is in-distribution to model what it is not, we instead propose to directly leverage the information at hand about the data.

---

**Algorithm 1** Pseudo-Input Generator (PIG)

---

**Input:** $p_{\text{data}}(\text{x})$, number of PIGs $K$, tolerance, max_iterations
**Draw initial PIGs:** $\forall k \in [1, K]$, $\hat{\text{x}}_k \sim p_{\text{data}}(\text{x})$
**Initialize:** $iterations = 0, \epsilon = \infty$
**repeat**
    compute $\forall k \in [1, K], \nabla_{\text{x}} p(\text{x})(\hat{\text{x}}_k)$
    $\epsilon = \max_{k \in [1,K]}(\|\delta \nabla_{\text{x}} p(\text{x})(\hat{\text{x}}_k)\|_2)$
    $\forall k \in [1, K], \hat{\text{x}}_k = \hat{\text{x}}_k - \delta \nabla_{\text{x}} p(\text{x})(\hat{\text{x}}_k)$
    iterations $+ = 1$
**until** (iterations < max_iterations) & ($\epsilon$ > tolerance)

---

Algorithm 1 generates pseudo-inputs with simple steps using the data density. Pseudo-inputs are initially drawn from $p_{\text{data}}(\text{x})$, and their positions iteratively updated with gradient descent to minimise their likelihood under $p_{\text{data}}(\text{x})$, similarly to reversed adversarial steps (Goodfellow et al., 2014b). In practice, we see little sensitivity of the model uncertainty on the chosen hyperparameters and on the number of pseudo-inputs[3].

Remarkably, this modular procedure can run prior to training, in parallel for all $\hat{\text{x}}_k$ with automatic differentiation, and thus results in limited additional complexity for the optimisation[4]. It relies on the availability of a differentiable density estimate of the data, which is, depending on the use case, either directly available (see Sec. 3.2), or can be approximated through a variety of methods such as *Bayesian Gaussian mixture models* (Bishop, 2006) (see Sec. 3.1). A caveat here is that depending on the PIG's parameters, and on the quality of the density estimate available, pseudo-inputs might be generated in undesired regions of the input space, e.g uninformative density minima. In practice, we adopted conservative density estimates and did not observe any significant degradation of the predictive uncertainty.

## 3 Experiments

**Holistic evaluation of uncertainty estimates**. The ground truth for uncertainty is usually unknown, making its evaluation non-trivial. As in Stirn & Knowles (2020), we propose to assess it using multiple metrics. Calibration, which evaluates probabilistic predictions w.r.t the long-run frequencies that actually occur (Dawid, 1982) can be measured by *proper scoring rules* (Lakshminarayanan et al., 2017) such as the model log-likelihood $\log p_\theta(\text{x}|\lambda)$. Additionally, the *root mean squared error (RMSE)* between the predictive and empirical variance, $\text{Var}[\text{x}] - (\mu_\theta(\text{x}) - \text{x})^2$, quantifies the model's awareness of its own uncertainty. It nevertheless requires an understanding of the model's mean predictive performance, as commonly measured by the RMSE of the mean residuals, $\mu_\theta(\text{x}) - \text{x}$. We further evaluate the cooperation of mean and uncertainty estimates for generating credible samples, which constitutes a consistency check for the learned precision distribution (Gelman et al., 2013), by measuring the RMSE of sample residuals $\text{x}^* - \text{x}$, with $\text{x}^* \sim p_\theta(\text{x})$. Finally, The ELBO, despite the absence of theoretical grounding for it (Blei et al., 2017), is commonly reported as an approximation of the marginal likelihood, and thus of the model's predictive performance.

A complete assessment of a model's uncertainty further requires its evaluation under distributional shift (Ovadia et al., 2019), which we introduce voluntarily through deliberate splitting of the training set (Sec. 3.1), or by using test data from a different dataset (Sec. 3.2).

### 3.1 Regression

In a regression setting where the proposed model must capture the conditioning $y \,|\, x$, the precision $\lambda$ of a Gaussian likelihood is the only assumed latent code.

---

[3]Sec. II..4 of the supplements.
[4]Running times are reported in Sec. V.

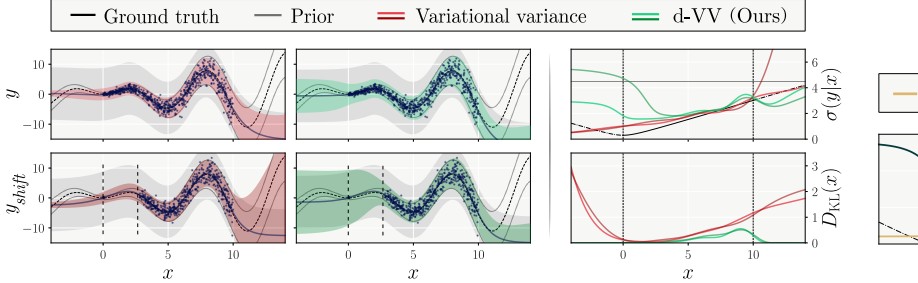 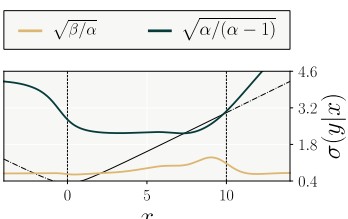

Figure 5: Toy regression results. On the left, are shown mean $\pm$ 2 std, with the training data of the bottom row presenting a shift (gap indicated by dashed lines). On the right are displayed the predictive uncertainty fit and the prior KL divergence.

Figure 6: Aleatoric (yellow) and epistemic (dark) uncertainties.

Faithfully to variational variance (Stirn & Knowles, 2020) we adopt a Gamma heteroscedastic variational posterior $q_\phi(\lambda|x) = \Gamma\left(\lambda|\alpha_\phi(x), \beta_\phi(x)\right)$ parametrised by the independent $\alpha_\phi$ and $\beta_\phi$ networks, with weights $\phi$, uniquely conditioned on the inputs (see Fig. 4). This approximate posterior, independent of the targets, gives up on the dependency of the true posterior on both covariates to guarantee heteroscedasticity[5].

For more than 2 degrees of freedom, corresponding to $\alpha_\phi(x) > 1$, the resulting marginal likelihood $p_{\theta,\phi}(y|x) = T\left(y \,|\, 2\alpha_\phi(x), \mu_\theta(x), \sqrt{\beta_\phi(x)/\alpha_\phi(x)}\right)$, has its first two moments defined, $\mathbb{E}[y|x] = \mu_\theta(x)$ and $\mathrm{Var}[y|x] = \beta_\phi(x)/(\alpha_\phi(x)-1)$, providing explicit mean and uncertainty estimates with a single forward pass in the $\alpha_\phi$, $\beta_\phi$ and $\mu_\theta$ networks. To ensure definition of both the posterior and marginal distributions' variance, the parameter maps use a soft-plus on their last layer to ensure positivity, and the $\alpha_\phi$ network is further shifted by 1.

For pseudo-inputs generation, we estimate the input density prior to training with a Bayesian Gaussian mixture model (Bishop, 2006). We refer to it henceforth as *dissipative variational variance (d-VV)*. The implementation details are listed in Sec. III. of the appendix.

### 3.1.1 Toy regression

The desiderata for our method are clear: capture of the data heteroscedasticity, extrapolation to a higher uncertainty level, no underestimation of the predictive uncertainty, and posterior extrapolation to the prior out-of-distribution. Skafte et al. (2019) first showed on the toy regression task, $y = x\sin(x) + 0.3\,\epsilon_1 + 0.3\,x\,\epsilon_2$, where $\epsilon_1, \epsilon_2 \sim \mathcal{N}(0,1)$, that amongst a collection of methods, only their proposed variance network architecture could realise our first three expectations. Fig. 5 demonstrates that our more principled approach also fulfills all of our requirements, without the need for arbitrarily enforcing the desired extrapolation in our architecture. The importance of out-of-distribution training is seen as the standard variational variance approach fails to produce uncertainty estimates that extrapolate correctly and are robust to distributional shift (bottom row of Fig. 5).

**Decomposing model and data uncertainty.** Fig. 6 shows that the aleatoric component captures the heteroscedastic increase of uncertainty in the training data while the epistemic uncertainty, constant in distribution, extrapolates to higher values.

### 3.1.2 UCI Benchmarks

Real world regression datasets from the UCI repository are used to evaluate our model against curated baselines, as in Hernández-Lobato & Adams (2015) and Skafte et al. (2019)[6]. We further copy each dataset with a distributional shift to assess the robustness of the methods. As in Foong et al. (2019), a shift is introduced, for each input feature, as a hole in the training data by assigning the middle third of observations to the test set, when sorted w.r.t that feature.

---

[5]The true posterior is in Sec. III. of the supplements.
[6]See Sec. III. for benchmark specification.

Table 1: Each cell counts datasets for which each method demonstrated the best average, over 5 trials. Grey shows statistical draws and "n/a" metrics impossible to evaluate for a method. Best per metric is highlighted in bold.

| UCI benchmarks (shifts included) | d-VV | VV | VV (no prior) | Mean variance network | Skafte et al | Deep ensembles | Monte Carlo dropout | Noise contrastive priors | Bayes by backprop |
|---|---|---|---|---|---|---|---|---|---|
| $\mathcal{L}$ | **21** / 18 | 3 / 3 | 0 / 0 | n / a | n / a | n / a | n / a | n / a | n / a |
| $\log p(y\|x)$ | 2 / 7 | 5 / 9 | 0 / 0 | 0 / 0 | 4 / 9 | 0 / 0 | 0 / 0 | 2 / 8 | **11** / 8 |
| $\mathrm{RMSE}[y, \mu(x)]$ | 2 / 2 | 1 / 4 | 2 / 5 | 1 / 5 | 2 / 3 | 5 / 12 | 3 / 16 | 0 / 0 | **8** / 0 |
| $\mathrm{RMSE}[\mathrm{Var}]$ | 2 / 4 | 5 / 9 | 2 / 4 | 0 / 0 | n / a | 2 / 5 | 5 / 17 | 0 / 0 | **8** / 7 |
| $\mathrm{RMSE}[y, \tilde{y}]$ | 3 / 4 | 5 / 7 | 4 / 2 | n / a | n / a | n / a | n / a | 1 / 2 | **11** / 7 |
| $\mathbb{E}[\mathrm{KL}]$ | **23** / 14 | 1 / 0 | 0 / 0 | n / a | n / a | n / a | n / a | n / a | n / a |

Table 1 aggregates best performances over the 24 datasets for two classes of methods[7]. To the left are methods that directly parametrise model uncertainty with neural networks, and to the right are methods that use ensembling for uncertainty estimation.

We note that *Bayes by backprop* offer a strong baseline, outperforming other ensemble methods. The instability of its performance on some datasets[7], as well as Hafner et al. (2018)'s demonstration of its overfitting in an active learning setting, challenge the reliability of its uncertainty estimates. As expected, the regularisation by NCP degrades significantly the mean predictions.

Our method performs best in its class. The ELBO reveals that it clearly strengthens the prior regularisation as introduced in VV, without significant degradation of the predictive power, as shown by other metrics. A caveat here is that the calibration of d-VV, as measured by the log-likelihood, suffers from the increased uncertainty of the method, which we deem acceptable if it leads to safer uncertainty estimates.

### 3.2 Generative models

We extend the evaluation of our proposal to the case of generative models through the lens of VAEs (Kingma & Welling, 2013; Rezende et al., 2014). VAEs infer a low dimensional latent encoding of the data $z \in \mathbb{R}^D$, on which is conditioned the generative process $p_\theta(\mathrm{x}|z)$. Its predictive uncertainty, which evaluates the confidence of the model in its ability to adequately reconstruct inputs is known to be untrustworthy.

In the case of continuous or seemingly continuous inputs, the adoption of a Gaussian decoder $p_\theta(\mathrm{x}|z) = \mathcal{N}\left(\mathrm{x}|\mu_\theta(z), \sigma_\theta^2(z)\right)$ results in an ill-defined model likelihood (Mattei & Frellsen, 2018b) that encourages decoder variance collapse, making the training of the model notoriously harder (Skafte et al., 2019). Most implementations therefore choose to fix the variance to a set level e.g $\sigma_\theta(z) = 0.1$, or elude the challenge by adopting a Bernoulli likelihood.

Motivated by our previous results, we now aim to demonstrate that VAEs, whose decoder is fitted with our method, are able to provide robust uncertainty estimates. Assuming a latent generative precision, the latent variables of the model are decomposed into $\mathrm{z} = \{z, \lambda\}$, with $z$ the latent input representations. The marginalisation of the Gamma distributed latent variance results in a Student-T decoder, as detailed in Eq. 1. The *variational variance variational auto encoder (V3AE)* yields, with the addition of our OOD pseudo-inputs training, the dissipative loss function[8],

$$\mathrm{L}(q_\phi, \theta; \mathcal{D}_{\mathrm{train}}) = - \sum_{\mathrm{x} \in \mathcal{D}_{\mathrm{train}}} \mathcal{L}(q_\phi, \theta; \mathrm{x})$$
$$+ \mathbb{E}_{q_{\mathrm{out}}(z)} \left[ D_{\mathrm{KL}}\left( q_\phi(\lambda|z) \,||\, p(\lambda) \right) \right]. \tag{6}$$

Because only the decoder is regularised, the pseudo-inputs lie in the space of latent representations, $\mathcal{D}_{\mathrm{out}} = \{\hat{z}_k\}_{k=1}^K \in \mathbb{R}^D$. The distribution of training inputs is therefore readily accessible as the aggre-

---

[7]Full results are included in Sec. III. of the supplements.
[8]The derivation is provided in Sec. IV.

Table 2: Evaluation of the generative modelling. For each dataset, we report mean $\pm$ std over 5 trials.

|  |  | FashionMNIST | SVHN | CIFAR |
|---|---|---|---|---|
| $\log p(\mathrm{x})$ | VAE | 2215.54±68.81 | **4304.90±58.45** | **2930.64±14.82** |
|  | d-V3AE | **2349.71±11.80** | 4133.41±64.28 | 2668.85±13.23 |
| $\mathrm{RMSE}(\mathrm{x}, \tilde{\mathrm{x}})$ | VAE | 0.171±0.003 | 0.097±7e-4 | 0.154±5e-4 |
|  | d-V3AE | **0.158±0.003** | **0.087±0.002** | **0.129±7e-4** |

gate posterior $q_\phi(z|\mathcal{D}_{\mathrm{train}}) \approx \frac{1}{N}\sum_{n=1}^{N} q_\phi(z|\mathrm{x}_n)$. Again, we rely on a split training procedure to leverage this perk; the encoder parameter maps $\mu_\theta$ and $\sigma_\theta$, as well as the decoder mean $\mu_\phi$ are first trained until convergence, allowing the generation of the OOD pseudo-inputs and subsequently, the training of the decoder variance.

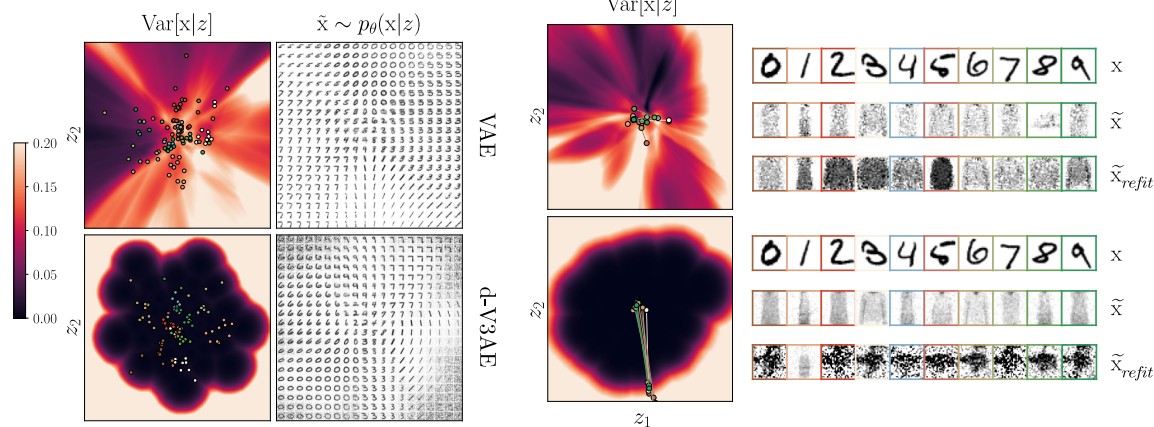

Figure 7: Decoder's aggregated variance (left) and generated samples (right) from the latent space. Coloured points correspond to latent representations of test data, with per-class colours.

Figure 8: Effect of encoder refitting on the latent representations (left) and resulting samples (right). OOD inputs (top rows, x) initially result in in-distribution samples (second rows, $\tilde{\mathrm{x}}$). The refitted encoder displaces the encodings (coloured trajectories), modifying the generated samples (third rows, $\tilde{\mathrm{x}}_{\mathrm{refit}}$).

**Image data.** We evaluate the performance of our proposed *dissipative V3AE (d-V3AE)* against a fully Gaussian VAE on image data, coming from FashionMNIST, SVHN and CIFAR10. For both models, all parameter maps share the same underlying architecture, with the addition of either a softplus and/or a shifting last layer to ensure definition of both the variational and the generative distribution's moments[9].

Table 2 compares model performance on two metrics, the log-likelihood and the RMSE between the original inputs x and reconstructed samples $\tilde{\mathrm{x}}$, where $\tilde{\mathrm{x}} \sim p_\theta(\mathrm{x}|\lambda, z), (\lambda, z) \sim q_\phi(\lambda, z|\mathrm{x})$. Unlike most previous implementations, we focus on actual samples, and not the mean, of the generative distributions. This comparison emphasize the cooperation between the decoder's mean and variance, allowing evaluation of the models' uncertainty estimates. Our method both qualitatively (Fig. 9), and quantitatively improves on a Gaussian VAE's sampling ability. The prior smoothens the uncertainty estimates, resulting in more realistic and less crisp samples. The log-likelihoods, evaluated at test time using truncation, i.e. $p_{\mathrm{trunc}}(\mathrm{x}) = p_\theta(\mathrm{x})/(F_{\mathrm{x}}(1) - F_{\mathrm{x}}(0))$, to account for the finite support of data, reveal that our model can achieve a better fit, if the prior is selected correctly. In SVHN and CIFAR10, the presence of color channels complicates the selection process and challenges our choice of a single homoscedastic prior for all pixels and channels. We note that the dissipative loss also applies to classic VAEs with Bernoulli-only decoders; see Sec. IV. of the supplements for details.

**Applications of robust generative uncertainty.** In Figs. 7 & 8, the colouring of the 2D latent space represent the aggregated decoder variance $\sum_{i=1}^{d}(\sigma_\theta^2(z))_i$. It is clear that our method displays more regular

---

[9]Full details are included in Sec. IV. of the supplements.

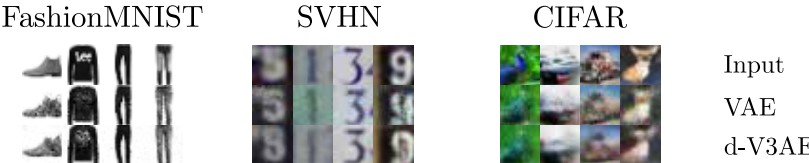

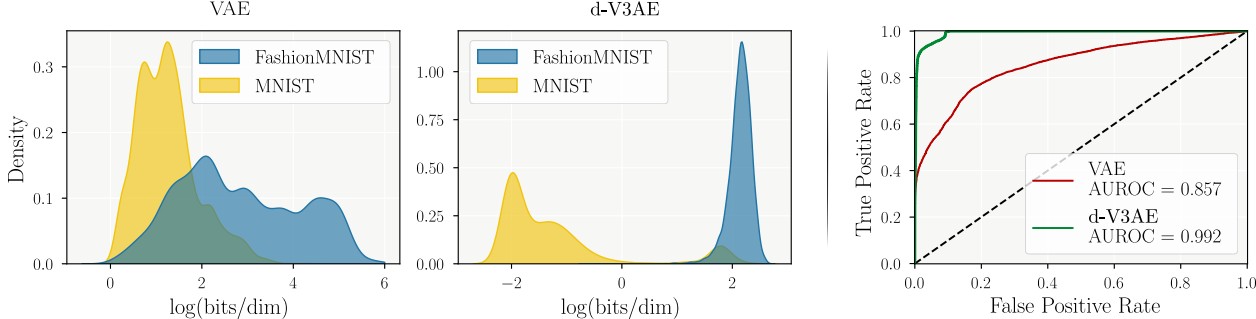

Figure 9: Generated samples $\tilde{x} \sim p_\theta(x|\lambda, z)$. The samples from d-V3AE are less noisy than those of the standard VAE.

Figure 10: Empirical densities of likelihoods for FashionMNIST (ID) and MNIST (OOD). The clear separation of distributions offered by our method is reflected in the high AUROC shown on the right. The d-V3AE beats Havtorn et al. (2021)'s reported AUROC of 0.984 .

uncertainty estimates, and provides the extrapolation guarantees we strove for. Beyond increased robustness and better generative power, this unlocks meaningful out-of-distribution detection, beating previous state-of-the-art (Havtorn et al., 2021)[10]. For Figs. 8 & 10, as argued in Mattei & Frellsen (2018a), we refit at test time the encoder of models trained on FashionMNIST on MNIST. The regularity and structure of the decoder variance rewards the encoder for learning to place representations of OOD data outside of the region of in-distribution latent encodings, resulting in a model that is aware of its own inability to reconstruct plausible data, as displayed by the row $\tilde{x}_{\text{refit}}$ of d-V3AE.

## 4 Conclusion

We have introduced a novel loss, the dissipative loss, that leverages artificial out-of-distribution pseudo-inputs for learning robust uncertainty estimates. We demonstrate through a Bayesian approach that casts a prior distribution over the model's variance a principled mechanism for controlling the extrapolation properties of neural networks governing the predictive uncertainty. Our results reflect the benefits of our principled and scalable approach, displaying better calibrated and more robust uncertainty estimates, while matching the predictive power of known baselines. Finally, and most interestingly, our approach can instill into probabilistic models a notion of their own ignorance, increasing their ability to *know what they don't know.*

The main limitation of our approach is that it depends on an input density estimate. In our experience, even coarse-grained densities are sufficient to significantly improve upon current approaches. However, as one rarely have guaranteed good estimates of the input density, our method cannot be approached as a black-box. One exception seems to be the application to VAEs, where the aggregated posterior, in our experience, always provide a suitable density estimate.

**Broader Impact Statement**

In this optional section, TMLR encourages authors to discuss possible repercussions of their work, notably any potential negative impact that a user of this research should be aware of. Authors should consult the TMLR Ethics Guidelines available on the TMLR website for guidance on how to approach this subject.

---

[10]Additional experiments are provided in Sec. IV..6

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

## I. Student-t likelihood

### I..1 Marginal distribution of a Gaussian likelihood with a Gamma precision

In the case of a Gaussian likelihood with a latent Gamma distributed precision, the marginal distribution follows:

$$
\begin{aligned}
p_\theta(\mathrm{x}) &= \int \mathcal{N}(\mathrm{x}|\mu, \lambda)\Gamma(\lambda|\alpha, \beta)d\lambda \\
&= \int \frac{\lambda^{1/2}}{\sqrt{2\pi}}e^{-\frac{1}{2}\lambda(\mathrm{x}-\mu)^2}\frac{\beta^\alpha}{\Gamma(\alpha)}\lambda^{\alpha-1}e^{-\beta\lambda}d\lambda \\
&= \frac{1}{\Gamma(\alpha)\sqrt{2\pi}}\frac{\beta^\alpha}{\left(\beta + \frac{(\mathrm{x}-\mu)^2}{2}\right)^{\alpha-\frac{1}{2}}}\int \left[\left(\beta + \frac{1}{2}(\mathrm{x}-\mu)^2\right)\lambda\right]^{(\alpha+\frac{1}{2})-1}e^{-\left(\beta + \frac{(\mathrm{x}-\mu)^2}{2}\right)\lambda}d\lambda \\
&= \frac{\Gamma\left(\alpha + \frac{1}{2}\right)}{\Gamma(\alpha)\sqrt{2\pi}}\frac{\beta^\alpha}{\left(\beta + \frac{(\mathrm{x}-\mu)^2}{2}\right)^{\alpha-\frac{1}{2}}} \\
&= \frac{\Gamma\left(\frac{2\alpha+1}{2}\right)}{\Gamma(\alpha)\sqrt{\pi 2\alpha}\left(\frac{\beta}{\alpha}\right)^{1/2}}\left(1 + \frac{1}{2\alpha}\left(\frac{\mathrm{x}-\mu}{\left(\frac{\beta}{\alpha}\right)^{1/2}}\right)^2\right)^{-\frac{2\alpha+1}{2}} \\
&= \frac{\Gamma\left(\frac{\nu+1}{2}\right)}{\Gamma\left(\frac{\nu}{2}\right)\sqrt{\nu\pi}\hat\sigma}\left(1 + \frac{1}{\nu}\left(\frac{\mathrm{x}-\hat\mu}{\hat\sigma}\right)^2\right)^{-\frac{\nu+1}{2}} \\
&= T\left(\mathrm{x}|\nu = 2\alpha, \hat\mu = \mu, \hat\sigma = \sqrt{\beta/\alpha}\right).
\end{aligned}
\tag{7}
$$

The moments of the marginal distribution are, assuming $\nu > 2$,

$$
\begin{cases}
\mathbb{E}[\mathrm{x}] &= \hat\mu = \mu \\
\mathrm{Var}[\mathrm{x}] &= \hat\sigma^2\frac{\nu}{\nu-2} = \frac{\beta}{\alpha}\frac{\alpha}{\alpha-1}.
\end{cases}
\tag{8}
$$

### I..2 Decomposition of a Student-t's uncertainty

The variance of a non standard Student-t distribution with $2\alpha$ degrees of freedom and scaled by $\sqrt{\beta/\alpha}$ can be decomposed as $\mathrm{Var}[\mathrm{x}] = \frac{\beta}{\alpha}\frac{\alpha}{\alpha-1}$. The number of degrees of freedom scales with the number of observations that the distribution arises from. When the number of observations grows towards $\infty$, i.e towards perfect information, the term $\frac{\alpha}{\alpha-1}$ converges towards 1, motivating its casting into an epistemic factor. The natural consequence is that $\frac{\beta}{\alpha}$, which accounts for the rest of the model's uncertainty, scales as the aleatoric uncertainty. For more details see Jørgensen (2020, p16).

## II. Pseudo-inputs generator

### II..1 On boundary samples

Tab. 3 provides supporting evidence regarding the benefits of placing pseudo-inputs close to the training distribution. It compares the mean measured OOD KL divergence over 5 trials for two different UCI regression datasets. The boundary OOD pseudo-inputs were generated using the proposed density-based generation procedure. The far OOD pseudo-inputs were generated by adding large Gaussian noise ($\sigma = 15$ for standardised inputs) to training points. Boundary OOD pseudo-inputs result in a lower OOD prior KL divergence, meaning that the regularisation is indeed enforced in most of the OOD support.

Table 3: Comparison of the OOD prior KL for different pseudo-input distributions. Lower is better, and best per experiment are highlighted in bold.

| method | CCPP | Wine-white |
|---|---|---|
| No OOD (VV) | 0.655 | 1.719 |
| Boundary OOD (d-VV) | **0.019** | **0.072** |
| Far OOD (d-VV) | 0.023 | 0.941 |

Table 4: Evaluation of the influence of the number of steps in Alg. 1.

| N steps | ELBO | $\log p(y|x)$ |
|---|---|---|
| 0 | **1.253** | 1.360 |
| 1 | 1.232 | 1.361 |
| 5 | 1.192 | 1.423 |
| 10 | 1.183 | **1.437** |

Table 5: Evaluation of the influence of the threshold $\epsilon$ in Alg. 1.

| Threshold $\epsilon$ | ELBO | $\log p(y|x)$ |
|---|---|---|
| 0.1 | 1.193 | **1.418** |
| 0.01 | 1.192 | 1.418 |
| 0.001 | **1.252** | 1.359 |

## II..2 Sensitivity to hyperparameters

The pseudo-input generator hyperparameters were chosen through a coarse grid search on a sub selection of experiments from the UCI benchmarks, and applied to the rest of the experiments. In practice we observed that Alg. 1 is relatively insensitive to hyperparameters. For example the Tab. 4 below describes the evolution of the ELBO and the log-likelihood as a function of the number of steps on the *Carbon* UCI dataset. The ELBO and LLK here have opposite evolutions based on the number of steps - hence our practical choice of a "middle ground" with n = 5. Tab. 5 evaluates similarly the influence of the threshold parameter. The results are naturally varying depending on the dataset chosen, as different overall input densities will more or less allow the gradient descent iterations to modify the original distribution of noisy pseudo-inputs. We regard this as a strength of our proposed method, if the distribution of pseudo-inputs can be improved and better separated from the training input distribution, it will leverage this possibility, but if it's not the case, pseudo inputs will not vary from their original position, generated as if they were inputs perturbed by Gaussian noise.

## II..3 Influence of the pseudo-input generator

We originally envisioned to generate pseudo-inputs using the same Gaussian perturbation technique as independently proposed in Hafner et al. (2018), $\hat{x} = x + \epsilon$ where $\epsilon \sim \mathcal{N}(0, \sigma^2)$. We nevertheless quickly come to realise that this does not actually produce pseudo-inputs that are guaranteed to be out-of-distribution, and eventually result in over-regularisation. Tab. 6 compares the mean ELBO over 5 trials of our proposal with selected parameters for the gradient descent (dVV), with our proposal without any gradient descent (dVV - 0 steps) and our method with pseudo-inputs generated as Gaussian noise (dVV - Gaussian noise, as is done in Hafner et al. (2018)) on the UCI benchmark. We observe that, as expected, better pseudo-inputs placement results in our method performing better on a larger selection of experiments (8/12).

Table 6: Comparison of the mean ELBO of dissipative VV on UCI datasets, over 5 trials, depending on the chosen pseudo-input generator. Best is highlighted in bold.

| method | Boston | Carbon | CCPP | Concrete | Energy | Kin8nm | Naval | Protein | Superconduct | Wine-red | Wine-white | Yacht |
|---|---|---|---|---|---|---|---|---|---|---|---|---|
| d-VV | **-0.614** | 1.191 | -0.141 | **-0.443** | **0.665** | **-0.247** | **0.522** | -1.319 | **-0.543** | -1.997 | -1.822 | -17.599 |
| d-VV (0 steps) | -0.658 | **1.229** | **-0.131** | **-0.433** | 0.656 | -0.255 | 0.496 | -1.317 | -0.558 | -2.192 | -2.03 | -21.802 |
| d-VV (Gaussian noise) | -0.819 | -0.473 | -0.569 | -0.645 | -0.515 | -0.591 | -0.537 | **-1.236** | -0.724 | **-1.747** | **-1.554** | **-0.507** |

### II..4  Number of pseudo-inputs

In our implementation, to stabilise training, we use the expected value over each dataset (in or out-of-distribution) by dividing each term by the number of data points used to compute them. This results in a very limited sensibility of our practical implementation in the number of pseudo-inputs used.

## III.  Regression experiments

### III..1  Variational Variance's ELBO Closed Form

For a Gaussian likelihood and a Gamma posterior, both terms of the ELBO have a closed form solution. Firstly the expected log-likelihood verifies:

$$
\begin{aligned}
\mathbb{E}_{q(\lambda|x)}\left[\log p(y|x,\lambda)\right] &= \int \log \mathcal{N}(y|\mu(x),\lambda)\Gamma\left(\lambda|\alpha(x),\beta(x)\right)d\lambda \\
&= \int -\frac{1}{2}\left(\log 2\pi - \log\lambda + \lambda(y-\mu(x))^2\right)\Gamma\left(\lambda|\alpha(x),\beta(x)\right)d\lambda \quad (9) \\
&= -\frac{1}{2}\left(\log 2\pi - \mathbb{E}_{q(\lambda|x)}[\log\lambda] + (y-\mu(x))^2\,\mathbb{E}_{q(\lambda|x)}[\lambda]\right).
\end{aligned}
$$

The variational posterior being Gamma distributed, its expected value is defined as $\mathbb{E}_{q(\lambda|x)}[\lambda] = \frac{\alpha(x)}{\beta(x)}$. The logarithmic expectation of a Gamma distribution can be derived to yield (Johnson et al., 1994, 337–349) $\mathbb{E}_{q(\lambda|x)}[\log\lambda] = \psi(\alpha(x)) - \log\beta(x)$ where $\psi$ is the digamma function. The closed-form expression of the expected likelihood is therefore:

$$
\mathbb{E}_{q(\lambda|x)}\left[\log p(y|x,\lambda)\right] = -\frac{1}{2}\left(\log 2\pi - \psi(\alpha(x)) + \log\beta(x) + \frac{\alpha(x)}{\beta(x)}(y-\mu(x))^2\right). \quad (10)
$$

Secondly, the KL-divergence between the posterior $\Gamma\left(\alpha(x),\beta(x)\right)$ and the prior $\Gamma(a,b)$ can be derived from Equation (28) in Bauckhage (2014, p6). With Bauckhage's notation, setting $p_1 = p_2 = 1$, to correspond to standard Gamma distributions, shape parameters $d_1 = \alpha(x)$ and $d_2 = a$, and scale parameters $a_1 = \frac{1}{\beta(x)}$ and $a_2 = \frac{1}{b}$ the KL-divergence can be expressed as

$$
\begin{aligned}
D_{\mathrm{KL}}(q(\lambda|x)\,||\,p(\lambda)) &= (\alpha(x)-a)\psi(\alpha(x)) \\
&\quad - \log\Gamma(\alpha(x)) + \log\Gamma(a) \\
&\quad + a(\log\beta(x) - \log b) \\
&\quad + \alpha(x)\frac{b-\beta(x)}{\beta(x)}.
\end{aligned} \quad (11)
$$

### III..2  True posterior and heteroscedasticity

The true posterior for variational variance in a regression context can be written $p(\lambda|y,x)$. As first demonstrated in Sec. 8.2 of Stirn & Knowles (2020), it factorizes as:

$$
p(\lambda|y,x) = \frac{p(y|x,\lambda)p(\lambda)}{\int p(y|x,\lambda)p(\lambda)d\lambda} \quad (12)
$$

$$
= \frac{\Pi_{n=1}^N p(y_n|x_n,\lambda_n)p(\lambda_n)}{\int \Pi_{n=1}^N p(y_n|x_n,\lambda_n)p(\lambda_n)d\lambda_n} \quad (13)
$$

$$
= \Pi_{n=1}^N \frac{p(y_n|x_n,\lambda_n)p(\lambda_n)}{\int p(y_n|x_n,\lambda_n)p(\lambda_n)d\lambda_n} \quad (14)
$$

$$
= \Pi_{n=1}^N p(\lambda_n|y_n,x_n). \quad (15)
$$

As a result, the true posterior both depends on the inputs $x_n$ and targets $y_n$. It means that a single input, could theoretically imply different latent precisions for different targets $y_n \neq y_k$, thus violating the x-surjectivity of the heteroscedastic definition.

### III..3 Model architecture

We adopted a unified network architecture for the regression case. All neural-network parameter maps share the same underlying architecture, a single hidden layer with 50 hidden units using *exponential linear unit (ELU)* activation functions. A final *softplus* layer is applied on the last layer of the $\sigma$, $\alpha$ and $\beta$ parameter maps. The $\alpha$ parameter map is further shifted by $+1$ to ensure the definition of the marginal distribution's variance. Regression models are trained with the *Adam* (Kingma & Ba, 2014) optimiser, and both the inputs and targets are standardised prior to training and testing.

### III..4 Pseudo-input generator

Tab. 7 presents the parameters used by the PIG in a regression setting. We remind that these parameters are parameters of a gradient descent, with learning rate $\delta$. For our experiments, we approximated the input

Table 7: Parameters for the regression pseudo-input generator.

| K | max_iterations | tolerance | $\delta$ |
|---|---|---|---|
| N | 5 | 0.005 | $4e$-1 |

density with a Bayesian Gaussian mixture model[11] with diagonal covariance matrices, and initialised with as many components as there are inputs in a batch.

### III..5 UCI experiments

Table 8: UCI benchmarks

| Name | Dimensions $(N, D_x, D_y)$ | Link (https://archive.ics.uci.edu/ml/*) |
|---|---|---|
| Boston | (505,13,1) | machine-learning-databases/housing/ |
| Carbon | (10721,5,3) | datasets/Carbon+Nanotubes |
| Concrete | (1030,8,1) | datasets/Concrete+Compressive+Strength |
| Energy | (768,8,2) | datasets/Energy+efficiency |
| Kin8nm | (8192,8,1) | https://www.openml.org/d/189 |
| Naval | (11934,16,2) | datasets/Condition+Based+Maintenance+of+Naval+Propulsion+Plants |
| Power plant (CCPP) | (9568,4,1) | datasets/Combined+Cycle+Power+Plant |
| Protein | (45630, 9, 1) | datasets/Physicochemical+Properties+of+Protein+Tertiary+Structure |
| Superconductivity | (21263,81,1) | datasets/Superconductivty+Data |
| Wine-red | (1599,11,1) | datasets/Wine+Quality |
| Wine-white | (4898,11,1) | datasets/Wine+Quality |
| Yacht | (308,6,1) | datasets/Yacht+Hydrodynamics |

The UCI experiments (https://archive.ics.uci.edu/ml/datasets.php) consist of the datasets presented in Tab. 8.

The results, for the different metrics, as presented in Tab. 11 to 16, were computed as the mean $\pm$ the standard deviation over 5 trials with standardised inputs and targets. Due to a technical error, we were forced to re-run the experiments for *d-VV* and *VV (no PIG)* right before the submission deadline, and reduced the number of trials to 3 for these methods.

A method is deemed to perform best for a given metric when the mean of the evaluated metric is the best across methods. For determining statistical draws we ran for each method a two-sided test for verifying whether the mean of the evaluated metric $\mu$ is significantly different to the mean of the best method $\mu_{\text{best}}$. To do so, we test for $\mu_{\text{best}} - \mu = 0$, under a Gaussian distribution with standard deviation $\frac{\sigma_{\text{best}}^2}{N_{\text{best}}} + \frac{\sigma^2}{N}$ at a 0.05 level.

---

[11]https://scikit-learn.org/stable/modules/generated/sklearn.mixture.BayesianGaussianMixture.html

### III..6  Aggregate benchmark data

Table 9: Results for experiments **excluding distributional shifts**. Each cell counts datasets for which each method demonstrated the best average, over 5 trials. Grey shows statistical draws and "n/a" metrics impossible to evaluate for a method. Best per metric is highlighted in bold.

| UCI benchmarks (shifts not included) | d-VV | VV | VV (no prior) | Mean variance network | Skafte et al | Deep ensembles | Monte Carlo dropout | Noise contrastive priors | Bayes by backprop |
|---|---|---|---|---|---|---|---|---|---|
| $\mathcal{L}$ | **11** / 8 | 1 / 1 | 0 / 0 | n / a | n / a | n / a | n / a | n / a | n / a |
| $\log p(y\|x)$ | 1 / 1 | 3 / 6 | 0 / 0 | 0 / 0 | 1 / 2 | 0 / 0 | 0 / 0 | 0 / 0 | **7** / 6 |
| RMSE$[y, \mu(x)]$ | 2 / 2 | 1 / 4 | 1 / 3 | 0 / 0 | 0 / 0 | **4** / 10 | 0 / 0 | 0 / 0 | 4 / 0 |
| RMSE[Var] | 2 / 4 | 3 / 4 | 1 / 2 | 0 / 0 | n / a | 1 / 3 | 1 / 2 | 0 / 0 | **4** / 4 |
| RMSE$[y, \tilde{y}]$ | 2 / 2 | 3 / 3 | 2 / 1 | n / a | n / a | n / a | n / a | 0 / 0 | **5** / 3 |
| $\mathbb{E}[\text{KL}]$ | **12** / 7 | 0 / 0 | 0 / 0 | n / a | n / a | n / a | n / a | n / a | n / a |

Table 10: Results for experiments **specifically for distributional shifts**. Each cell counts datasets for which each method demonstrated the best average, over 5 trials. Grey shows statistical draws and "n/a" metrics impossible to evaluate for a method. Best per metric is highlighted in bold.

| UCI benchmarks (only shifts) | d-VV | VV | VV (no prior) | Mean variance network | Skafte et al | Deep ensembles | Monte Carlo dropout | Noise contrastive priors | Bayes by backprop |
|---|---|---|---|---|---|---|---|---|---|
| $\mathcal{L}$ | **10** / 10 | 2 / 2 | 0 / 0 | n / a | n / a | n / a | n / a | n / a | n / a |
| $\log p(y\|x)$ | 1 / 6 | 2 / 3 | 0 / 0 | 0 / 0 | 3 / 7 | 0 / 0 | 0 / 0 | 2 / 8 | **4** / 2 |
| RMSE$[y, \mu(x)]$ | 0 / 0 | 0 / 0 | 1 / 2 | 1 / 5 | 2 / 3 | 1 / 2 | 3 / 16 | 0 / 0 | **4** / 0 |
| RMSE[Var] | 0 / 0 | 2 / 5 | 1 / 2 | 0 / 0 | n / a | 1 / 2 | **4** / 15 | 0 / 0 | 4 / 3 |
| RMSE$[y, \tilde{y}]$ | 1 / 2 | 2 / 4 | 2 / 1 | n / a | n / a | n / a | n / a | 1 / 2 | **6** / 4 |
| $\mathbb{E}[\text{KL}]$ | **11** / 7 | 1 / 0 | 0 / 0 | n / a | n / a | n / a | n / a | n / a | n / a |

### III..7 Benchmark raw data

Table 11: UCI benchmarks - $\mathcal{L}$

| | UCI benchmarks | d-VV | VV | VV (no prior) | Mean variance network | Skafte et al | Deep ensembles | Monte Carlo dropout | Noise contrastive priors | Bayes by backprop |
|---|---|---|---|---|---|---|---|---|---|---|
| Not shifted | uci_boston | -0.61 ± 0.33 | -0.72 ± 0.38 | -64.28 ± 29.34 | nan ± nan | nan ± nan | nan ± nan | nan ± nan | nan ± nan | nan ± nan |
| | uci_carbon | 1.19 ± 0.11 | 1.17 ± 0.12 | -3913.5 ± 580.03 | nan ± nan | nan ± nan | nan ± nan | nan ± nan | nan ± nan | nan ± nan |
| | uci_ccpp | -0.14 ± 0.01 | -0.16 ± 0.04 | -10.9 ± 1.06 | nan ± nan | nan ± nan | nan ± nan | nan ± nan | nan ± nan | nan ± nan |
| | uci_concrete | -0.44 ± 0.13 | -0.46 ± 0.08 | -44.06 ± 14.23 | nan ± nan | nan ± nan | nan ± nan | nan ± nan | nan ± nan | nan ± nan |
| | uci_energy | 0.67 ± 0.03 | 0.65 ± 0.03 | -118.78 ± 81.29 | nan ± nan | nan ± nan | nan ± nan | nan ± nan | nan ± nan | nan ± nan |
| | uci_kin8nm | -0.25 ± 0.02 | -0.28 ± 0.04 | -16.76 ± 1.53 | nan ± nan | nan ± nan | nan ± nan | nan ± nan | nan ± nan | nan ± nan |
| | uci_naval | 0.52 ± 0.16 | 0.12 ± 0.4 | -9.57 ± 4.57 | nan ± nan | nan ± nan | nan ± nan | nan ± nan | nan ± nan | nan ± nan |
| | uci_protein | -1.32 ± 0.01 | -1.34 ± 0.01 | -14.31 ± 2.59 | nan ± nan | nan ± nan | nan ± nan | nan ± nan | nan ± nan | nan ± nan |
| | uci_superconduct | -0.54 ± 0.03 | -0.56 ± 0.01 | -269.08 ± 58.88 | nan ± nan | nan ± nan | nan ± nan | nan ± nan | nan ± nan | nan ± nan |
| | uci_wine_red | -2.0 ± 0.08 | -2.36 ± 0.19 | -37.86 ± 31.48 | nan ± nan | nan ± nan | nan ± nan | nan ± nan | nan ± nan | nan ± nan |
| | uci_wine_white | -1.82 ± 0.06 | -2.01 ± 0.1 | -869.26 ± 1718.81 | nan ± nan | nan ± nan | nan ± nan | nan ± nan | nan ± nan | nan ± nan |
| | uci_yacht | -17.6 ± 0.43 | 1.04 ± 0.1 | -551.12 ± 1060.39 | nan ± nan | nan ± nan | nan ± nan | nan ± nan | nan ± nan | nan ± nan |
| Shifted | uci_boston | -1.28 ± 0.32 | -1.47 ± 0.28 | -110.63 ± 110.55 | nan ± nan | nan ± nan | nan ± nan | nan ± nan | nan ± nan | nan ± nan |
| | uci_carbon | 1.16 ± 0.02 | 1.11 ± 0.03 | -3991.86 ± 679.64 | nan ± nan | nan ± nan | nan ± nan | nan ± nan | nan ± nan | nan ± nan |
| | uci_ccpp | -0.25 ± 0.08 | -0.33 ± 0.14 | -8.36 ± 1.65 | nan ± nan | nan ± nan | nan ± nan | nan ± nan | nan ± nan | nan ± nan |
| | uci_concrete | -1.29 ± 0.27 | -1.51 ± 0.38 | -75.74 ± 59.7 | nan ± nan | nan ± nan | nan ± nan | nan ± nan | nan ± nan | nan ± nan |
| | uci_energy | -0.69 ± 1.1 | -0.56 ± 0.9 | -891.92 ± 1570.09 | nan ± nan | nan ± nan | nan ± nan | nan ± nan | nan ± nan | nan ± nan |
| | uci_kin8nm | -0.33 ± 0.03 | -0.38 ± 0.04 | -23.9 ± 1.92 | nan ± nan | nan ± nan | nan ± nan | nan ± nan | nan ± nan | nan ± nan |
| | uci_naval | -4.73 ± 4.61 | -5.31 ± 6.58 | -26.02 ± 48.06 | nan ± nan | nan ± nan | nan ± nan | nan ± nan | nan ± nan | nan ± nan |
| | uci_protein | -1.56 ± 0.11 | -1.64 ± 0.16 | -9.34 ± 6.68 | nan ± nan | nan ± nan | nan ± nan | nan ± nan | nan ± nan | nan ± nan |
| | uci_superconduct | -1.43 ± 0.32 | -1.52 ± 0.34 | -224.0 ± 111.57 | nan ± nan | nan ± nan | nan ± nan | nan ± nan | nan ± nan | nan ± nan |
| | uci_wine_red | -2.84 ± 0.21 | -3.45 ± 0.35 | -47.78 ± 56.02 | nan ± nan | nan ± nan | nan ± nan | nan ± nan | nan ± nan | nan ± nan |
| | uci_wine_white | -2.04 ± 0.11 | -2.53 ± 0.29 | -346.83 ± 434.27 | nan ± nan | nan ± nan | nan ± nan | nan ± nan | nan ± nan | nan ± nan |
| | uci_yacht | -7.59 ± 2.58 | 0.35 ± 0.19 | -194.42 ± 149.45 | nan ± nan | nan ± nan | nan ± nan | nan ± nan | nan ± nan | nan ± nan |

Table 12: UCI benchmarks - $\log p(y|x)$

| | UCI benchmarks | d-VV | VV | VV (no prior) | Mean variance network | Skafte et al | Deep ensembles | Monte Carlo dropout | Noise contrastive priors | Bayes by backprop |
|---|---|---|---|---|---|---|---|---|---|---|
| Not shifted | uci_boston | -0.43 ± 0.35 | -0.42 ± 0.39 | -3.34 ± 1.39 | -0.76 ± 0.07 | -0.18 ± 0.19 | -0.68 ± 0.04 | -0.81 ± 0.51 | -1.39 ± 0.33 | -248.43 ± 163.55 |
| | uci_carbon | 1.45 ± 0.13 | 1.45 ± 0.11 | 0.98 ± 3.08 | -3.78 ± 0.05 | 1.13 ± 0.51 | -3.71 ± 0.04 | 0.29 ± 1.08 | nan ± nan | nan ± nan |
| | uci_ccpp | -0.07 ± 0.01 | -0.03 ± 0.03 | 0.05 ± 0.06 | -0.58 ± 0.14 | -0.18 ± 0.12 | -0.61 ± 0.05 | -3.36 ± 0.56 | 0.21 ± 0.04 | 4.06 ± 0.69 |
| | uci_concrete | -0.29 ± 0.15 | -0.25 ± 0.1 | -0.83 ± 0.5 | -0.68 ± 0.09 | -0.4 ± 0.15 | -0.65 ± 0.04 | -0.9 ± 0.33 | 0.38 ± 0.04 | 3.84 ± 0.66 |
| | uci_energy | 0.87 ± 0.04 | 0.89 ± 0.03 | 0.47 ± 0.2 | -1.22 ± 0.11 | 0.28 ± 0.37 | -1.17 ± 0.04 | 0.36 ± 0.26 | nan ± nan | nan ± nan |
| | uci_kin8nm | -0.17 ± 0.02 | -0.15 ± 0.04 | -0.36 ± 0.07 | -0.61 ± 0.06 | -0.61 ± 0.12 | -0.65 ± 0.03 | -0.63 ± 0.05 | -0.68 ± 0.08 | -0.16 ± 0.03 |
| | uci_naval | 0.71 ± 0.19 | 0.52 ± 0.2 | -0.13 ± 0.32 | -2.26 ± 0.08 | -2.67 ± 0.22 | -2.26 ± 0.06 | -0.2 ± 0.74 | nan ± nan | nan ± nan |
| | uci_protein | -1.16 ± 0.01 | -1.12 ± 0.01 | -1.42 ± 0.38 | -1.13 ± 0.05 | -1.54 ± 0.74 | -1.05 ± 0.01 | -7.41 ± 0.27 | -1.02 ± 0.01 | -0.96 ± 0.02 |
| | uci_superconduct | -0.38 ± 0.02 | -0.35 ± 0.02 | -1.73 ± 0.25 | -0.96 ± 0.18 | -0.68 ± 0.03 | -1.72 ± 0.25 | -0.2 ± 0.19 | -0.04 ± 0.06 | |
| | uci_wine_red | -1.91 ± 0.07 | -2.13 ± 0.21 | -7.77 ± 6.39 | -2560.95 ± 5395.69 | -1.15 ± 0.04 | -1.24 ± 0.08 | -4.24 ± 0.91 | 0.16 ± 0.04 | 3.76 ± 0.39 |
| | uci_wine_white | -1.72 ± 0.06 | -1.75 ± 0.1 | -305.7 ± 549.73 | -27.69 ± 48.8 | -1.4 ± 0.58 | -1.16 ± 0.08 | -5.86 ± 1.08 | 0.29 ± 0.06 | 3.76 ± 0.82 |
| | uci_yacht | 0.9 ± 0.02 | 1.33 ± 0.11 | 0.63 ± 0.59 | -0.59 ± 0.11 | 0.4 ± 0.14 | -0.58 ± 0.04 | 0.33 ± 0.69 | 0.63 ± 0.1 | 1.57 ± 0.6 |
| Shifted | uci_boston | -1.09 ± 0.32 | -1.16 ± 0.27 | -10.33 ± 10.87 | -0.84 ± 0.09 | -0.16 ± 0.09 | -0.79 ± 0.07 | -2.52 ± 1.34 | -3.83 ± 1.82 | -428.29 ± 194.72 |
| | uci_carbon | 1.34 ± 0.06 | 1.4 ± 0.02 | -0.6 ± 2.9 | -3.87 ± 0.34 | 1.12 ± 0.25 | -3.71 ± 0.12 | 0.55 ± 0.1 | nan ± nan | nan ± nan |
| | uci_ccpp | -0.18 ± 0.09 | -0.2 ± 0.14 | -0.1 ± 0.08 | -0.54 ± 0.04 | -0.16 ± 0.02 | -0.65 ± 0.02 | -4.31 ± 0.43 | 0.2 ± 0.07 | 3.76 ± 0.75 |
| | uci_concrete | -1.11 ± 0.27 | -1.23 ± 0.36 | -10.81 ± 15.27 | -0.77 ± 0.1 | -0.38 ± 0.06 | -0.78 ± 0.04 | -2.38 ± 0.52 | 0.29 ± 0.05 | 4.17 ± 0.36 |
| | uci_energy | -0.47 ± 1.15 | -0.2 ± 0.76 | -96.95 ± 259.19 | -1.42 ± 0.26 | 0.2 ± 0.25 | -1.36 ± 0.23 | -1.13 ± 2.52 | nan ± nan | nan ± nan |
| | uci_kin8nm | -0.26 ± 0.04 | -0.22 ± 0.04 | -0.88 ± 0.14 | -0.64 ± 0.09 | -0.59 ± 0.04 | -0.65 ± 0.05 | -0.86 ± 0.21 | -0.68 ± 0.09 | -0.26 ± 0.1 |
| | uci_naval | -4.55 ± 4.62 | -5.01 ± 6.53 | -14.78 ± 18.92 | -3.59 ± 0.89 | -2.76 ± 0.15 | -3.62 ± 0.89 | -22.29 ± 12.05 | nan ± nan | nan ± nan |
| | uci_protein | -1.44 ± 0.12 | -1.42 ± 0.15 | -2.26 ± 1.3 | -1.33 ± 0.09 | -1.51 ± 0.46 | -1.21 ± 0.07 | -9.94 ± 1.27 | -1.16 ± 0.06 | -1.21 ± 0.09 |
| | uci_superconduct | -1.28 ± 0.32 | -1.28 ± 0.33 | -8.32 ± 11.59 | -0.98 ± 0.11 | -1.06 ± 0.16 | -0.9 ± 0.06 | -5.9 ± 1.74 | -0.72 ± 0.28 | -2.87 ± 5.54 |
| | uci_wine_red | -2.7 ± 0.2 | -3.1 ± 0.32 | -12.13 ± 13.11 | -229.56 ± 749.24 | -1.14 ± 0.02 | -1.57 ± 0.26 | -4.42 ± 0.5 | 0.13 ± 0.1 | 3.84 ± 0.44 |
| | uci_wine_white | -1.97 ± 0.12 | -2.24 ± 0.27 | -183.07 ± 323.25 | -1.75 ± 0.23 | -1.3 ± 0.13 | -1.29 ± 0.04 | -5.65 ± 0.49 | 0.3 ± 0.04 | 3.61 ± 0.88 |
| | uci_yacht | 0.71 ± 0.76 | 0.65 ± 0.19 | 0.51 ± 0.55 | -0.53 ± 0.05 | 0.43 ± 0.07 | -0.58 ± 0.05 | 0.37 ± 0.35 | 0.41 ± 0.2 | -11.62 ± 25.18 |

Table 13: UCI benchmarks - RMSE $[y, \mu(x)]$

| | UCI benchmarks | d-VV | VV | VV (no prior) | Mean variance network | Skafte et al | Deep ensembles | Monte Carlo dropout | Noise contrastive priors | Bayes by backprop |
|---|---|---|---|---|---|---|---|---|---|---|
| Not shifted | uci_boston | $0.33 \pm 0.09$ | $0.33 \pm 0.08$ | $0.38 \pm 0.09$ | $0.35 \pm 0.06$ | $0.3 \pm 0.07$ | $0.29 \pm 0.05$ | $0.33 \pm 0.05$ | $0.47 \pm 0.06$ | $0.5 \pm 0.04$ |
| | uci_carbon | $0.03 \pm 0.02$ | $0.03 \pm 0.02$ | $0.03 \pm 0.02$ | $0.75 \pm 0.01$ | $0.09 \pm 0.08$ | $0.75 \pm 0.01$ | $0.08 \pm 0.0$ | nan $\pm$ nan | nan $\pm$ nan |
| | uci_ccpp | $0.23 \pm 0.01$ | $0.23 \pm 0.01$ | $0.23 \pm 0.01$ | $0.23 \pm 0.01$ | $0.27 \pm 0.03$ | $0.23 \pm 0.01$ | $0.24 \pm 0.01$ | $0.07 \pm 0.03$ | $0.0 \pm 0.0$ |
| | uci_concrete | $0.29 \pm 0.04$ | $0.29 \pm 0.04$ | $0.33 \pm 0.04$ | $0.29 \pm 0.01$ | $0.35 \pm 0.07$ | $0.27 \pm 0.01$ | $0.28 \pm 0.02$ | $0.08 \pm 0.02$ | $0.0 \pm 0.0$ |
| | uci_energy | $0.08 \pm 0.01$ | $0.08 \pm 0.01$ | $0.3 \pm 0.03$ | $0.13 \pm 0.01$ | $0.22 \pm 0.08$ | $0.13 \pm 0.01$ | $0.13 \pm 0.02$ | nan $\pm$ nan | nan $\pm$ nan |
| | uci_kin8nm | $0.26 \pm 0.01$ | $0.26 \pm 0.01$ | $0.28 \pm 0.01$ | $0.27 \pm 0.01$ | $0.44 \pm 0.07$ | $0.26 \pm 0.01$ | $0.33 \pm 0.01$ | $0.4 \pm 0.07$ | $0.29 \pm 0.0$ |
| | uci_naval | $0.09 \pm 0.06$ | $0.1 \pm 0.03$ | $0.33 \pm 0.07$ | $0.72 \pm 0.01$ | $0.86 \pm 0.09$ | $0.72 \pm 0.01$ | $0.2 \pm 0.07$ | nan $\pm$ nan | nan $\pm$ nan |
| | uci_protein | $0.71 \pm 0.01$ | $0.71 \pm 0.0$ | $0.75 \pm 0.01$ | $0.71 \pm 0.01$ | $1.12 \pm 0.73$ | $0.69 \pm 0.01$ | $0.7 \pm 0.01$ | $0.76 \pm 0.02$ | $0.73 \pm 0.01$ |
| | uci_superconduct | $0.35 \pm 0.01$ | $0.35 \pm 0.01$ | $0.4 \pm 0.02$ | $0.35 \pm 0.01$ | $0.67 \pm 0.22$ | $0.32 \pm 0.01$ | $0.33 \pm 0.01$ | $0.44 \pm 0.03$ | $0.41 \pm 0.01$ |
| | uci_wine_red | $0.89 \pm 0.01$ | $0.9 \pm 0.04$ | $0.77 \pm 0.07$ | $1.13 \pm 0.15$ | $0.76 \pm 0.02$ | $0.84 \pm 0.06$ | $0.77 \pm 0.05$ | $0.1 \pm 0.04$ | $0.01 \pm 0.0$ |
| | uci_wine_white | $0.9 \pm 0.03$ | $0.88 \pm 0.02$ | $0.82 \pm 0.06$ | $0.85 \pm 0.05$ | $0.93 \pm 0.39$ | $0.77 \pm 0.06$ | $0.79 \pm 0.05$ | $0.08 \pm 0.04$ | $0.01 \pm 0.0$ |
| | uci_yacht | $0.05 \pm 0.02$ | $0.04 \pm 0.02$ | $0.82 \pm 0.12$ | $0.05 \pm 0.01$ | $0.09 \pm 0.06$ | $0.05 \pm 0.01$ | $0.11 \pm 0.04$ | $0.09 \pm 0.02$ | $0.16 \pm 0.03$ |
| Shifted | uci_boston | $0.52 \pm 0.08$ | $0.52 \pm 0.08$ | $0.43 \pm 0.1$ | $0.48 \pm 0.07$ | $0.3 \pm 0.04$ | $0.44 \pm 0.08$ | $0.41 \pm 0.06$ | $0.5 \pm 0.07$ | $0.46 \pm 0.06$ |
| | uci_carbon | $0.03 \pm 0.0$ | $0.03 \pm 0.0$ | $0.03 \pm 0.0$ | $0.72 \pm 0.05$ | $0.1 \pm 0.03$ | $0.72 \pm 0.05$ | $0.08 \pm 0.0$ | nan $\pm$ nan | nan $\pm$ nan |
| | uci_ccpp | $0.26 \pm 0.02$ | $0.26 \pm 0.03$ | $0.25 \pm 0.01$ | $0.25 \pm 0.01$ | $0.26 \pm 0.01$ | $0.25 \pm 0.01$ | $0.25 \pm 0.01$ | $0.07 \pm 0.04$ | $0.01 \pm 0.0$ |
| | uci_concrete | $0.54 \pm 0.07$ | $0.54 \pm 0.07$ | $0.44 \pm 0.05$ | $0.48 \pm 0.05$ | $0.35 \pm 0.03$ | $0.43 \pm 0.05$ | $0.4 \pm 0.04$ | $0.09 \pm 0.03$ | $0.0 \pm 0.0$ |
| | uci_energy | $0.26 \pm 0.23$ | $0.26 \pm 0.22$ | $0.39 \pm 0.14$ | $0.28 \pm 0.19$ | $0.23 \pm 0.05$ | $0.22 \pm 0.11$ | $0.22 \pm 0.11$ | nan $\pm$ nan | nan $\pm$ nan |
| | uci_kin8nm | $0.28 \pm 0.01$ | $0.28 \pm 0.01$ | $0.29 \pm 0.01$ | $0.29 \pm 0.01$ | $0.43 \pm 0.02$ | $0.27 \pm 0.01$ | $0.36 \pm 0.02$ | $0.37 \pm 0.08$ | $0.31 \pm 0.02$ |
| | uci_naval | $1.37 \pm 0.96$ | $1.35 \pm 0.95$ | $1.61 \pm 0.71$ | $0.89 \pm 0.11$ | $0.9 \pm 0.06$ | $0.89 \pm 0.1$ | $1.32 \pm 0.59$ | nan $\pm$ nan | nan $\pm$ nan |
| | uci_protein | $0.83 \pm 0.04$ | $0.83 \pm 0.05$ | $0.84 \pm 0.04$ | $0.84 \pm 0.05$ | $1.02 \pm 0.22$ | $0.8 \pm 0.05$ | $0.8 \pm 0.03$ | $0.82 \pm 0.03$ | $0.81 \pm 0.03$ |
| | uci_superconduct | $0.59 \pm 0.07$ | $0.59 \pm 0.06$ | $0.52 \pm 0.06$ | $0.6 \pm 0.06$ | $0.6 \pm 0.08$ | $0.52 \pm 0.05$ | $0.51 \pm 0.05$ | $0.54 \pm 0.06$ | $0.53 \pm 0.05$ |
| | uci_wine_red | $1.13 \pm 0.05$ | $1.14 \pm 0.05$ | $0.79 \pm 0.03$ | $1.62 \pm 0.21$ | $0.76 \pm 0.01$ | $1.03 \pm 0.11$ | $0.84 \pm 0.03$ | $0.11 \pm 0.05$ | $0.01 \pm 0.0$ |
| | uci_wine_white | $0.96 \pm 0.04$ | $0.96 \pm 0.04$ | $0.87 \pm 0.03$ | $1.03 \pm 0.07$ | $0.84 \pm 0.05$ | $0.86 \pm 0.03$ | $0.86 \pm 0.04$ | $0.06 \pm 0.02$ | $0.01 \pm 0.0$ |
| | uci_yacht | $0.16 \pm 0.08$ | $0.1 \pm 0.04$ | $0.74 \pm 0.22$ | $0.09 \pm 0.03$ | $0.07 \pm 0.02$ | $0.07 \pm 0.02$ | $0.13 \pm 0.04$ | $0.11 \pm 0.02$ | $0.18 \pm 0.06$ |

Table 14: UCI benchmarks - RMSE $\left[\mathrm{Var}[y|x], (y - \mu(x))^2\right]$

| | UCI benchmarks | d-VV | VV | VV (no prior) | Mean variance network | Skafte et al | Deep ensembles | Monte Carlo dropout | Noise contrastive priors | Bayes by backprop |
|---|---|---|---|---|---|---|---|---|---|---|
| Not shifted | uci_boston | $0.25 \pm 0.11$ | $0.37 \pm 0.26$ | $1.000000e{+}11 \pm 3.162278e{+}11$ | $0.61 \pm 0.14$ | nan $\pm$ nan | $0.53 \pm 0.06$ | $0.29 \pm 0.17$ | $36.0 \pm 4.18$ | $0.77 \pm 0.31$ |
| | uci_carbon | $0.05 \pm 0.04$ | $0.05 \pm 0.04$ | $0.03 \pm 0.03$ | $1.68 \pm 0.06$ | nan $\pm$ nan | $1.6 \pm 0.03$ | $0.09 \pm 0.01$ | nan $\pm$ nan | nan $\pm$ nan |
| | uci_ccpp | $0.15 \pm 0.01$ | $0.15 \pm 0.01$ | $0.24 \pm 0.3$ | $0.5 \pm 0.18$ | nan $\pm$ nan | $0.47 \pm 0.04$ | $0.14 \pm 0.04$ | $0.13 \pm 0.01$ | $0.0 \pm 0.0$ |
| | uci_concrete | $0.24 \pm 0.15$ | $0.22 \pm 0.13$ | $1.32 \pm 3.53$ | $0.55 \pm 0.15$ | nan $\pm$ nan | $0.47 \pm 0.05$ | $0.18 \pm 0.03$ | $0.25 \pm 0.06$ | $0.0 \pm 0.0$ |
| | uci_energy | $0.03 \pm 0.0$ | $0.02 \pm 0.0$ | $0.15 \pm 0.04$ | $0.55 \pm 0.06$ | nan $\pm$ nan | $0.49 \pm 0.02$ | $0.08 \pm 0.01$ | nan $\pm$ nan | nan $\pm$ nan |
| | uci_kin8nm | $0.14 \pm 0.02$ | $0.13 \pm 0.02$ | $63.3 \pm 146.89$ | $0.48 \pm 0.09$ | nan $\pm$ nan | $0.47 \pm 0.04$ | $0.2 \pm 0.01$ | $0.38 \pm 0.03$ | $0.14 \pm 0.01$ |
| | uci_naval | $0.03 \pm 0.0$ | $3.828603e{+}03 \pm 6.631150e{+}03$ | $6.000000e{+}11 \pm 5.163978e{+}11$ | $2.0 \pm 0.2$ | nan $\pm$ nan | $1.93 \pm 0.16$ | $0.12 \pm 0.05$ | nan $\pm$ nan | nan $\pm$ nan |
| | uci_protein | $0.8 \pm 0.04$ | $0.8 \pm 0.01$ | $3.000000e{+}11 \pm 4.830458e{+}11$ | $0.88 \pm 0.09$ | nan $\pm$ nan | $0.77 \pm 0.03$ | $0.88 \pm 0.03$ | $9.76 \pm 3.17$ | $4.71 \pm 10.99$ |
| | uci_superconduct | $0.39 \pm 0.04$ | $0.41 \pm 0.07$ | $5.371845e{+}04 \pm 8.860245e{+}04$ | $0.53 \pm 0.08$ | nan $\pm$ nan | $0.53 \pm 0.05$ | $0.33 \pm 0.06$ | $0.62 \pm 0.08$ | $0.62 \pm 0.05$ |
| | uci_wine_red | $1.38 \pm 0.08$ | $1.62 \pm 0.38$ | $3.46 \pm 4.12$ | $2.69 \pm 0.78$ | nan $\pm$ nan | $2.97 \pm 2.35$ | $1.18 \pm 0.21$ | $0.57 \pm 0.17$ | $0.0 \pm 0.0$ |
| | uci_wine_white | $1.52 \pm 0.01$ | $1.52 \pm 0.14$ | $3.795848e{+}03 \pm 9.990566e{+}03$ | $1.5 \pm 0.33$ | nan $\pm$ nan | $1.11 \pm 0.2$ | $1.24 \pm 0.16$ | $0.08 \pm 0.01$ | $0.0 \pm 0.0$ |
| | uci_yacht | $0.03 \pm 0.0$ | $0.01 \pm 0.0$ | $2.272965e{+}03 \pm 4.416696e{+}03$ | $0.59 \pm 0.13$ | nan $\pm$ nan | $0.52 \pm 0.05$ | $0.09 \pm 0.03$ | $0.28 \pm 0.05$ | $0.03 \pm 0.02$ |
| Shifted | uci_boston | $0.74 \pm 0.37$ | $0.74 \pm 0.35$ | $7.692309e{+}10 \pm 2.773501e{+}11$ | $0.72 \pm 0.19$ | nan $\pm$ nan | $0.71 \pm 0.16$ | $0.46 \pm 0.21$ | $37.59 \pm 17.11$ | $0.87 \pm 0.35$ |
| | uci_carbon | $0.04 \pm 0.01$ | $0.04 \pm 0.01$ | $0.04 \pm 0.01$ | $1.31 \pm 0.51$ | nan $\pm$ nan | $1.45 \pm 0.22$ | $0.09 \pm 0.0$ | nan $\pm$ nan | nan $\pm$ nan |
| | uci_ccpp | $0.15 \pm 0.03$ | $0.15 \pm 0.03$ | $0.13 \pm 0.01$ | $0.4 \pm 0.06$ | nan $\pm$ nan | $0.49 \pm 0.02$ | $0.14 \pm 0.02$ | $0.12 \pm 0.01$ | $0.0 \pm 0.0$ |
| | uci_concrete | $0.61 \pm 0.2$ | $0.64 \pm 0.21$ | $26.51 \pm 73.79$ | $0.57 \pm 0.16$ | nan $\pm$ nan | $0.54 \pm 0.09$ | $0.31 \pm 0.06$ | $0.28 \pm 0.12$ | $0.0 \pm 0.0$ |
| | uci_energy | $0.28 \pm 0.29$ | $0.25 \pm 0.26$ | $119.45 \pm 271.5$ | $0.55 \pm 0.09$ | nan $\pm$ nan | $0.59 \pm 0.2$ | $0.12 \pm 0.08$ | nan $\pm$ nan | nan $\pm$ nan |
| | uci_kin8nm | $0.16 \pm 0.01$ | $0.15 \pm 0.02$ | $33.73 \pm 71.3$ | $0.49 \pm 0.1$ | nan $\pm$ nan | $0.46 \pm 0.06$ | $0.23 \pm 0.04$ | $0.41 \pm 0.07$ | $0.17 \pm 0.03$ |
| | uci_naval | $4.51 \pm 4.91$ | $4.17 \pm 4.9$ | $4.375000e{+}11 \pm 5.123475e{+}11$ | $29.35 \pm 35.96$ | nan $\pm$ nan | $30.74 \pm 37.17$ | $2.47 \pm 1.51$ | nan $\pm$ nan | nan $\pm$ nan |
| | uci_protein | $0.98 \pm 0.1$ | $1.03 \pm 0.12$ | $7.777778e{+}11 \pm 4.409585e{+}11$ | $1.02 \pm 0.08$ | nan $\pm$ nan | $0.88 \pm 0.09$ | $1.04 \pm 0.07$ | $7.09 \pm 2.31$ | $1.58 \pm 0.89$ |
| | uci_superconduct | $0.67 \pm 0.15$ | $0.66 \pm 0.14$ | $5.061728e{+}11 \pm 5.030769e{+}11$ | $0.73 \pm 0.11$ | nan $\pm$ nan | $0.7 \pm 0.1$ | $0.53 \pm 0.09$ | $0.57 \pm 0.1$ | $0.65 \pm 0.32$ |
| | uci_wine_red | $2.34 \pm 0.43$ | $2.37 \pm 0.45$ | $4.02 \pm 5.18$ | $4.88 \pm 1.35$ | nan $\pm$ nan | $7.66 \pm 6.9$ | $1.32 \pm 0.12$ | $0.45 \pm 0.29$ | $0.0 \pm 0.0$ |
| | uci_wine_white | $1.57 \pm 0.15$ | $1.63 \pm 0.16$ | $2.788480e{+}03 \pm 8.365328e{+}03$ | $1.83 \pm 0.32$ | nan $\pm$ nan | $1.35 \pm 0.15$ | $1.4 \pm 0.14$ | $0.13 \pm 0.09$ | $0.0 \pm 0.0$ |
| | uci_yacht | $0.14 \pm 0.09$ | $0.04 \pm 0.01$ | $7.139074e{+}04 \pm 1.741110e{+}05$ | $0.52 \pm 0.07$ | nan $\pm$ nan | $0.51 \pm 0.05$ | $0.09 \pm 0.04$ | $0.47 \pm 0.5$ | $0.05 \pm 0.03$ |

Table 15: UCI benchmarks - RMSE $[y, \tilde{y}]$

| | UCI benchmarks | d-VV | VV | VV (no prior) | Mean variance network | Skafte et al | Deep ensembles | Monte Carlo dropout | Noise contrastive priors | Bayes by backprop |
|---|---|---|---|---|---|---|---|---|---|---|
| Not shifted | uci_boston | 0.56 ± 0.08 | 0.57 ± 0.16 | 0.58 ± 0.28 | nan ± nan | nan ± nan | nan ± nan | nan ± nan | 2.23 ± 0.68 | 0.51 ± 0.06 |
| | uci_carbon | 0.11 ± 0.01 | 0.1 ± 0.0 | 0.04 ± 0.02 | nan ± nan | nan ± nan | nan ± nan | nan ± nan | nan ± nan | nan ± nan |
| | uci_ccpp | 0.42 ± 0.01 | 0.4 ± 0.01 | 0.33 ± 0.01 | nan ± nan | nan ± nan | nan ± nan | nan ± nan | 0.33 ± 0.01 | 0.01 ± 0.01 |
| | uci_concrete | 0.44 ± 0.06 | 0.43 ± 0.05 | 0.43 ± 0.05 | nan ± nan | nan ± nan | nan ± nan | nan ± nan | 0.32 ± 0.04 | 0.01 ± 0.0 |
| | uci_energy | 0.21 ± 0.03 | 0.17 ± 0.01 | 0.4 ± 0.08 | nan ± nan | nan ± nan | nan ± nan | nan ± nan | nan ± nan | nan ± nan |
| | uci_kin8nm | 0.46 ± 0.01 | 0.43 ± 0.01 | 0.39 ± 0.04 | nan ± nan | nan ± nan | nan ± nan | nan ± nan | 0.78 ± 0.06 | 0.4 ± 0.01 |
| | uci_naval | 0.22 ± 0.03 | 0.29 ± 0.03 | 1.3 ± 1.0 | nan ± nan | nan ± nan | nan ± nan | nan ± nan | nan ± nan | nan ± nan |
| | uci_protein | 1.05 ± 0.01 | 1.05 ± 0.02 | 1.15 ± 0.13 | nan ± nan | nan ± nan | nan ± nan | nan ± nan | 1.24 ± 0.15 | 1.17 ± 0.38 |
| | uci_superconduct | 0.54 ± 0.0 | 0.56 ± 0.02 | 0.59 ± 0.11 | nan ± nan | nan ± nan | nan ± nan | nan ± nan | 0.62 ± 0.04 | 0.6 ± 0.04 |
| | uci_wine_red | 1.15 ± 0.1 | 1.07 ± 0.06 | 1.07 ± 0.09 | nan ± nan | nan ± nan | nan ± nan | nan ± nan | 0.4 ± 0.05 | 0.01 ± 0.0 |
| | uci_wine_white | 1.19 ± 0.04 | 1.17 ± 0.06 | 1.18 ± 0.12 | nan ± nan | nan ± nan | nan ± nan | nan ± nan | 0.31 ± 0.01 | 0.01 ± 0.0 |
| | uci_yacht | 0.18 ± 0.03 | 0.1 ± 0.01 | 0.97 ± 0.22 | nan ± nan | nan ± nan | nan ± nan | nan ± nan | 0.33 ± 0.13 | 0.23 ± 0.13 |
| Shifted | uci_boston | 0.73 ± 0.07 | 0.67 ± 0.08 | 0.58 ± 0.2 | nan ± nan | nan ± nan | nan ± nan | nan ± nan | 3.24 ± 1.19 | 0.5 ± 0.07 |
| | uci_carbon | 0.12 ± 0.01 | 0.11 ± 0.0 | 0.04 ± 0.01 | nan ± nan | nan ± nan | nan ± nan | nan ± nan | nan ± nan | nan ± nan |
| | uci_ccpp | 0.44 ± 0.01 | 0.43 ± 0.02 | 0.35 ± 0.02 | nan ± nan | nan ± nan | nan ± nan | nan ± nan | 0.33 ± 0.03 | 0.01 ± 0.0 |
| | uci_concrete | 0.73 ± 0.07 | 0.71 ± 0.04 | 0.52 ± 0.06 | nan ± nan | nan ± nan | nan ± nan | nan ± nan | 0.37 ± 0.04 | 0.01 ± 0.0 |
| | uci_energy | 0.44 ± 0.16 | 0.45 ± 0.19 | 0.69 ± 0.49 | nan ± nan | nan ± nan | nan ± nan | nan ± nan | nan ± nan | nan ± nan |
| | uci_kin8nm | 0.48 ± 0.03 | 0.45 ± 0.01 | 0.4 ± 0.02 | nan ± nan | nan ± nan | nan ± nan | nan ± nan | 0.79 ± 0.07 | 0.41 ± 0.03 |
| | uci_naval | 1.73 ± 0.8 | 1.71 ± 0.82 | 2.33 ± 1.32 | nan ± nan | nan ± nan | nan ± nan | nan ± nan | nan ± nan | nan ± nan |
| | uci_protein | 1.15 ± 0.03 | 1.18 ± 0.06 | 1.25 ± 0.13 | nan ± nan | nan ± nan | nan ± nan | nan ± nan | 1.2 ± 0.05 | 1.13 ± 0.04 |
| | uci_superconduct | 0.79 ± 0.05 | 0.8 ± 0.06 | 0.93 ± 0.42 | nan ± nan | nan ± nan | nan ± nan | nan ± nan | 0.71 ± 0.08 | 0.72 ± 0.1 |
| | uci_wine_red | 1.32 ± 0.05 | 1.36 ± 0.08 | 1.03 ± 0.05 | nan ± nan | nan ± nan | nan ± nan | nan ± nan | 0.44 ± 0.06 | 0.01 ± 0.0 |
| | uci_wine_white | 1.24 ± 0.07 | 1.23 ± 0.03 | 1.21 ± 0.09 | nan ± nan | nan ± nan | nan ± nan | nan ± nan | 0.31 ± 0.02 | 0.01 ± 0.0 |
| | uci_yacht | 0.22 ± 0.09 | 0.21 ± 0.02 | 0.97 ± 0.28 | nan ± nan | nan ± nan | nan ± nan | nan ± nan | 0.48 ± 0.23 | 0.22 ± 0.08 |

Table 16: UCI benchmarks - $\mathbb{E}[\text{KL}]$

| | UCI benchmarks | d-VV | VV | VV (no prior) | Mean variance network | Skafte et al | Deep ensembles | Monte Carlo dropout | Noise contrastive priors | Bayes by backprop |
|---|---|---|---|---|---|---|---|---|---|---|
| Not shifted | uci_boston | 0.24 ± 0.05 | 0.66 ± 0.29 | 109.96 ± 65.06 | nan ± nan | nan ± nan | nan ± nan | nan ± nan | nan ± nan | nan ± nan |
| | uci_carbon | 0.03 ± 0.01 | 0.2 ± 0.01 | 3.590175e+03 ± 683.49 | nan ± nan | nan ± nan | nan ± nan | nan ± nan | nan ± nan | nan ± nan |
| | uci_ccpp | 0.02 ± 0.01 | 0.65 ± 0.15 | 765.25 ± 1.304725e+03 | nan ± nan | nan ± nan | nan ± nan | nan ± nan | nan ± nan | nan ± nan |
| | uci_concrete | 0.06 ± 0.01 | 0.54 ± 0.09 | 151.72 ± 197.18 | nan ± nan | nan ± nan | nan ± nan | nan ± nan | nan ± nan | nan ± nan |
| | uci_energy | 0.03 ± 0.01 | 1.22 ± 0.06 | 719.58 ± 835.96 | nan ± nan | nan ± nan | nan ± nan | nan ± nan | nan ± nan | nan ± nan |
| | uci_kin8nm | 0.01 ± 0.0 | 0.21 ± 0.03 | 25.55 ± 5.99 | nan ± nan | nan ± nan | nan ± nan | nan ± nan | nan ± nan | nan ± nan |
| | uci_naval | 0.57 ± 0.13 | 1.09 ± 0.37 | 966.35 ± 1.672143e+03 | nan ± nan | nan ± nan | nan ± nan | nan ± nan | nan ± nan | nan ± nan |
| | uci_protein | 0.03 ± 0.0 | 0.88 ± 0.3 | 1.489842e+09 ± 9.803508e+08 | nan ± nan | nan ± nan | nan ± nan | nan ± nan | nan ± nan | nan ± nan |
| | uci_superconduct | 0.1 ± 0.02 | 0.81 ± 0.01 | 5.670007e+03 ± 5.534678e+03 | nan ± nan | nan ± nan | nan ± nan | nan ± nan | nan ± nan | nan ± nan |
| | uci_wine_red | 0.07 ± 0.01 | 2.22 ± 1.01 | 3.798346e+03 ± 6.945584e+03 | nan ± nan | nan ± nan | nan ± nan | nan ± nan | nan ± nan | nan ± nan |
| | uci_wine_white | 0.07 ± 0.01 | 1.72 ± 0.2 | 1.766196e+06 ± 2.118534e+06 | nan ± nan | nan ± nan | nan ± nan | nan ± nan | nan ± nan | nan ± nan |
| | uci_yacht | 0.14 ± 0.04 | 0.5 ± 0.29 | 869.08 ± 1.882199e+03 | nan ± nan | nan ± nan | nan ± nan | nan ± nan | nan ± nan | nan ± nan |
| Shifted | uci_boston | 0.13 ± 0.03 | 0.39 ± 0.29 | 390.83 ± 741.65 | nan ± nan | nan ± nan | nan ± nan | nan ± nan | nan ± nan | nan ± nan |
| | uci_carbon | 0.03 ± 0.01 | 0.2 ± 0.02 | 3.145017e+03 ± 808.29 | nan ± nan | nan ± nan | nan ± nan | nan ± nan | nan ± nan | nan ± nan |
| | uci_ccpp | 0.02 ± 0.0 | 0.25 ± 0.07 | 29.45 ± 18.62 | nan ± nan | nan ± nan | nan ± nan | nan ± nan | nan ± nan | nan ± nan |
| | uci_concrete | 0.05 ± 0.01 | 0.17 ± 0.05 | 549.89 ± 1.020544e+03 | nan ± nan | nan ± nan | nan ± nan | nan ± nan | nan ± nan | nan ± nan |
| | uci_energy | 0.02 ± 0.01 | 0.67 ± 0.41 | 2.454811e+03 ± 1.613791e+03 | nan ± nan | nan ± nan | nan ± nan | nan ± nan | nan ± nan | nan ± nan |
| | uci_kin8nm | 0.01 ± 0.0 | 0.19 ± 0.01 | 43.01 ± 8.46 | nan ± nan | nan ± nan | nan ± nan | nan ± nan | nan ± nan | nan ± nan |
| | uci_naval | 0.13 ± 0.02 | 0.3 ± 0.11 | 3.223887e+04 ± 1.257591e+05 | nan ± nan | nan ± nan | nan ± nan | nan ± nan | nan ± nan | nan ± nan |
| | uci_protein | 0.04 ± 0.01 | 0.72 ± 0.24 | 1.800074e+08 ± 2.840555e+08 | nan ± nan | nan ± nan | nan ± nan | nan ± nan | nan ± nan | nan ± nan |
| | uci_superconduct | 0.03 ± 0.01 | 0.53 ± 0.11 | 2.629295e+03 ± 3.482120e+03 | nan ± nan | nan ± nan | nan ± nan | nan ± nan | nan ± nan | nan ± nan |
| | uci_wine_red | 0.06 ± 0.01 | 1.04 ± 0.31 | 210.6 ± 258.73 | nan ± nan | nan ± nan | nan ± nan | nan ± nan | nan ± nan | nan ± nan |
| | uci_wine_white | 0.08 ± 0.01 | 2.05 ± 0.31 | 3.506659e+05 ± 6.077032e+05 | nan ± nan | nan ± nan | nan ± nan | nan ± nan | nan ± nan | nan ± nan |
| | uci_yacht | 0.33 ± 0.09 | 0.19 ± 0.04 | 196.98 ± 125.57 | nan ± nan | nan ± nan | nan ± nan | nan ± nan | nan ± nan | nan ± nan |

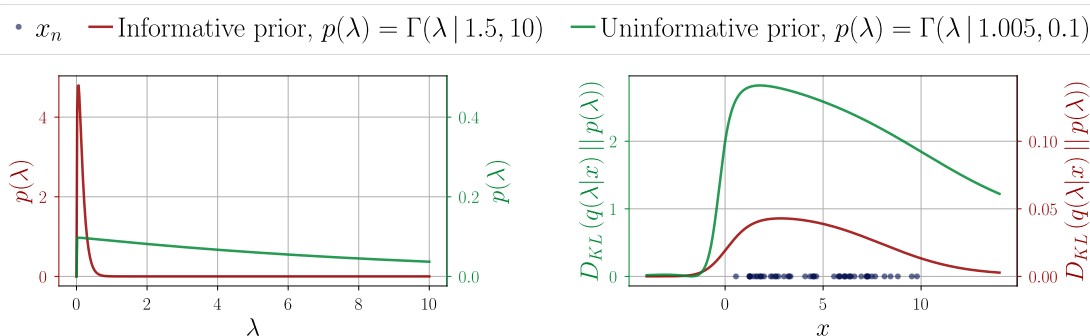

Figure 11: Effect of the informativity of the prior (as displayed on the left) on the KL divergence of the trained posterior on an artificial example (right). The scale of the respective KL divergences reveals that the heavy-tailed prior (green) allows the posterior to be significantly influenced by data, while its counterpart (red) is much more restrictive.

### III..8    Prior parameters

For the toy experiments (Fig. 3, 5 and 6), an homoscedastic prior that matches the standard deviation of the targets $\bar{\sigma}$ is chosen. As shown in Fig. 11, for a Gamma prior, the rate $\beta$ controls its informativity, the closer $\beta$ is to 0, the more spread out the prior is, and the less penalising it is for the posterior to diverge from it. We thus deliberately choose a prior with low informativity, $\beta = $ 1e-3, and infer the shape as $\alpha = 1 + \beta/\bar{\sigma}$. For the UCI benchmarks, we aimed to adopt a prior that would match the model's empirical variance $(y - \mu(x))^2$. As such, we first ran a training run to determine the model's empirical variance on each dataset, and subsequently adopted $\alpha = 1.5$ and $\beta = (\alpha - 1)(y - \mu(x))^2$, with the choice for $\alpha$ being motivated by stability concerns, and obtained from an empirical study. All prior parameters can be found in the configuration files present in the source code.

## IV.  Generative models experiments

Table 17: Datasets for generative models

| Name | Dimensions $(N, C, D_x, D_y)$ | Link |
|---|---|---|
| MNIST | $(70000, 1, 28, 28)$ | `http://yann.lecun.com/exdb/mnist/` |
| FashionMNIST | $(70000, 1, 28, 28)$ | `https://github.com/zalandoresearch/fashion-mnist` |
| EMNIST | $(70000, 1, 28, 28)$ | `https://www.westernsydney.edu.au/icns/reproducible_research/publication_support_materials/emnist` |
| KMNIST | $(70000, 1, 28, 28)$ | `https://github.com/rois-codh/kmnist` |
| SVHN | $(600000, 3, 32, 32)$ | `http://ufldl.stanford.edu/housenumbers/` |
| CIFAR | $(60000, 3, 32, 32)$ | `https://www.cs.toronto.edu/~kriz/cifar.html` |

### IV..1  Datasets

Tab. 17 lists all the datasets used in the generative modelling experiments.

### IV..2  Dissipative loss for generative models

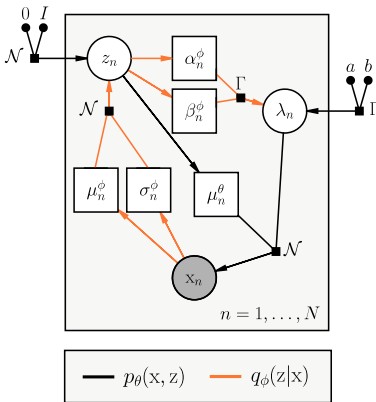

Figure 12: PGM for V3AE

The full expression of the dissipative loss for the V3AE is given as:

$$
\begin{aligned}
\text{Loss}(q_\phi, \theta; \mathcal{D}_{\text{train}}) = -\Big[ \Big( &\sum_{\text{x} \in \mathcal{D}_{\text{train}}} \mathbb{E}_{q_\phi(z|\text{x})} \big[ \mathbb{E}_{q_\phi(\lambda|z)} [\log p_\theta(\text{x}|z)] - D_{\text{KL}} \left( q_\phi(\lambda|z) \,||\, p(\lambda) \right) \big] \\
&- D_{\text{KL}}(q_\phi(z|\text{x}) \,||\, p(z)) \Big) + \mathbb{E}_{q_{\text{out}}(z)} \left[ D_{\text{KL}} \left( q_\phi(\lambda|z) \,||\, p(\lambda) \right) \right] \Big].
\end{aligned}
\tag{16}
$$

The expected likelihood w.r.t the posterior $q_\phi(z|\text{x})$ is intractable as it requires the integration of the parameter maps $\alpha_\phi(z)$ and $\beta_\phi(z)$ and must be approximated through MC-integration, using multiple sampled latent codes. We observed in practice that a low number of sampled codes, typically, 2 or 3, is sufficient for ensuring convergence.

### IV..3  Model architecture

In our VAEs, the encoder and decoder networks' architectures are mirrored, and all parameter maps of each stage share on the same architecture. For the MNIST and FashionMNIST datasets, we relied on fully connected encoder-decoders, with 2 hidden layers with respectively 512 and 256 neurons. Each fully connected layer is followed by *batch normalisation* (Ioffe & Szegedy, 2015). For the SVHN and CIFAR datasets, we relied on a convolutional architecture, with hidden dimensions corresponding to depths of 32, 64, 128, 256, and 512, for kernels of size 3, with a stride of 2 and a padding of 1. Again, batch normalisation is applied

after each layer. In both cases, we used *leaky rectified linear units (Leaky ReLU)* for activations and here again, softplus and shifting might be applied on the last layer of the different parameter maps to ensure the proper definition of the quantities they model, and models are optimised with Adam.

### IV..4  Pseudo-input generator

Table 18: Parameters for the generative model pseudo-input generator

| K | max_iterations | tolerance | $\delta$ |
|---|---|---|---|
| N | 10 | 0.007 | $4e$-1 |

Tab. 18 presents the parameters used by the PIG in a generative modelling setting. We remind that these parameters are parameters of a gradient descent, with learning rate $\delta$. Because it is too computationally expensive to use the complete aggregate posterior as the density estimate we base the PIG on, we iteratively generated pseudo-inputs using the aggregate posterior established on one batch at a time.

### IV..5  Prior parameters

As for the regression experiments, an homoscedastic Gamma prior, with the same parameters for all image channels was chosen for model comparisons. The shape and rate parameters were tuned with the same base intuition as for the UCI benchmarks; the prior uncertainty should be fairly close to the empirical mean of the model. An empirical grid search was conducted to determine the best combination of prior parameters wrt to the objective to optimise. In the case of out-of-distribution detection, we adopted an heteroscedastic prior. Such prior adopts similar base parameters as the more standard homoscedastic prior, but its rate parameter, and consequently its associated uncertainty, increases linearly as a function of the distance to the closest of the $C$ pre-determined K-means[12] cluster center, where $C$ is the number of classes in the dataset. Again, the prior parameters used for running the experiments can be found in the configuration files provided with the source code.

### IV..6  Out-of-distribution detection

The same experimental setup used to assess the OOD detection capabilities of the d-V3AE is replicated for other OOD datasets, namely *EMNIST* (Fig. 13) and *KMNIST* (Fig. 14). Here again, the benefits of the regularity of the learned decoder variance for OOD detection are clear, with our method clearly overperforming the baseline.

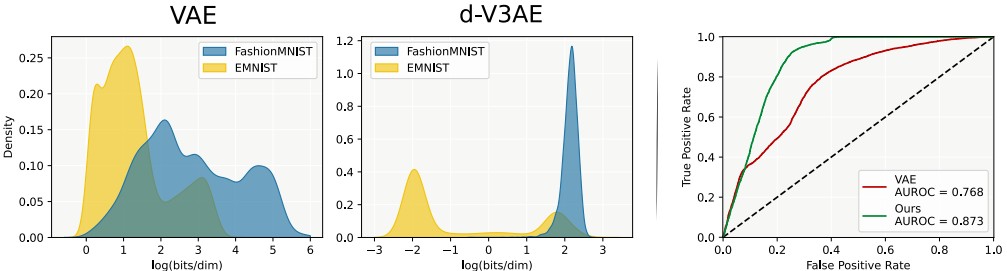

Figure 13: Empirical densities of likelihoods for FashionMNIST (ID) and EMNIST (OOD).

### IV..7  Pseudo-inputs training for VAE's with Bernoulli likelihood

Motivated by the idea of not necessarily having to adopt a non trivial $\Gamma(\lambda|a, b)$ prior, we explore the use of pseudo-input training in simpler VAE's with Bernoulli likelihood. In this setting, the combined epistemic and aleatoric uncertainty on the reconstructed $\tilde{x}$ is approximated with a measure of entropy. As uncertainty is high for distributions with high entropy, we reinterpret the decoded Bernoulli distributed reconstruction $\tilde{x}$ as

---

[12]https://scikit-learn.org/stable/modules/generated/sklearn.cluster.KMeans.html

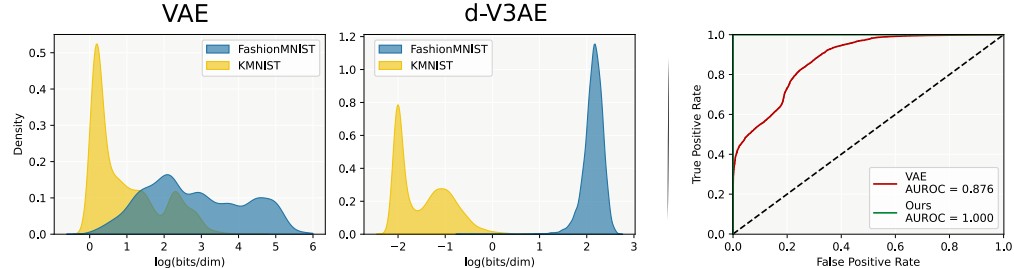

Figure 14: Empirical densities of likelihoods for FashionMNIST (ID) and KMNIST (OOD).

normalized Categorical distribution and then proceed to maximize its entropy for the pseudo inputs $\hat{z}$. The resulting loss function,

$$\text{Loss}(q_\phi, \theta; \mathcal{D}_{\text{train}}) = -\Big[ \sum_{\text{x} \in \mathcal{D}_{\text{train}}} \mathcal{L}(q_\phi, \theta; \text{x}) - \sum_{\hat{z} \in \mathcal{D}_{\text{out}}} \text{H}[\tilde{\text{x}}|\hat{z}] \Big], \tag{17}$$

balances the overall entropy of the reconstruction by promoting entropy increase, $\text{H}[\tilde{\text{x}}|\hat{z}]$. Figure 15 shows the effect of this method for a VAE trained on a subset of MNIST.

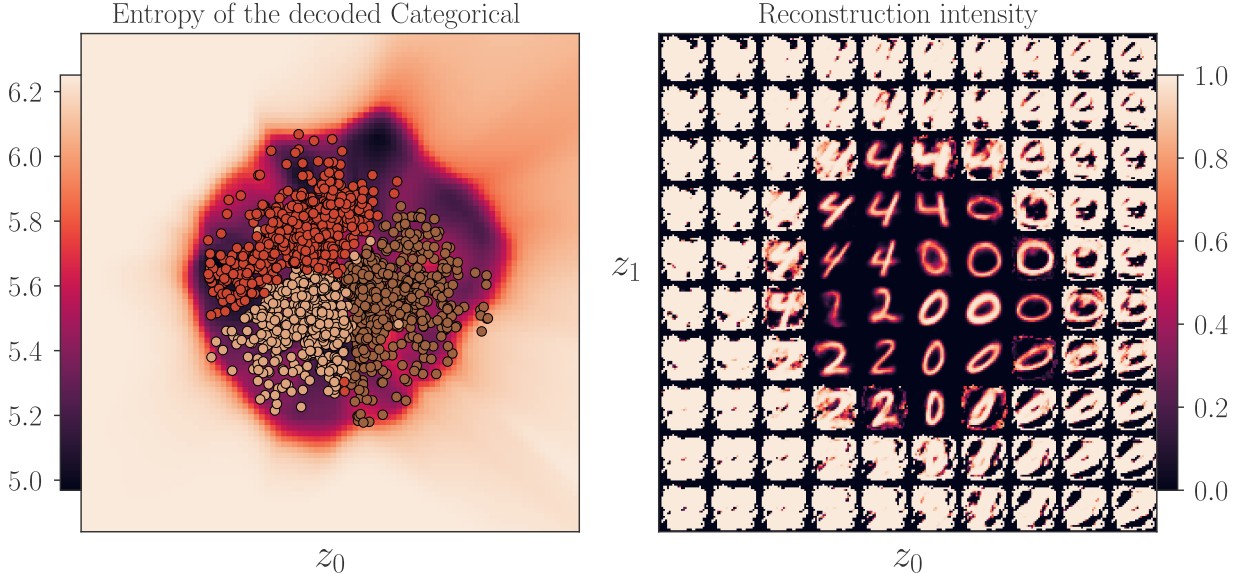

Figure 15: Pseudo inputs for Bernoulli VAE's

# V. Implementation details

### V..1 Source code

The source code is accessible in the GitHub repository `https://github.com/****`[13]

### V..2 Hardware

Experiments were done on a 16-Core AMD Ryzen 5950X machine with a single Nvidia 3080 GPU.

### V..3 Running times and carbon emissions

Table 19: Regression models

| Dataset | $CO_2$ (kg) | Time (s) |
|---|---|---|
| uci_boston | 0.000064 | 15.969 |
| uci_carbon | 0.000695 | 138.548 |
| uci_ccpp | 0.001464 | 275.824 |
| uci_concrete | 0.000102 | 22.480 |
| uci_energy | 0.000095 | 21.140 |
| uci_kin8nm | 0.000869 | 168.273 |
| uci_naval | 0.001427 | 274.547 |
| uci_protein | 0.009920 | 1762.071 |
| uci_superconduct | 0.002010 | 360.634 |
| uci_wine_red | 0.000185 | 36.186 |
| uci_wine_white | 0.000482 | 89.373 |
| uci_yacht | 0.000046 | 11.404 |

Table 20: Generative models

| Dataset | $CO_2$ (kg) | Time (s) |
|---|---|---|
| fashion_mnist | 0.003713 | 504.190 |
| cifar | 0.037554 | 2573.566 |
| svhn | 0.036692 | 2585.144 |

Table 21: Generative models w/o pseudo inputs

| Dataset | $CO_2$ (kg) | Time (s) |
|---|---|---|
| fashion_mnist | 0.002750 | 407.991 |
| cifar | 0.025522 | 2022.803 |
| svhn | 0.025130 | 2023.293 |

Tab. 20 and 21 demonstrate that the generation of artificial pseudo-inputs incurs a limited additional computational burden ($\sim +26\%$).

---

[13]Hidden for the review, please refer instead to the .zip folder attached.

