# OpenReview forum: "Robust uncertainty estimates with out-of-distribution pseudo-inputs training"
_TMLR — Rejected by TMLR_

### Review · Reviewer_obnL · 2022-08-15

**Summary Of Contributions:**

The authors tackle the problem that neural network-based generative approaches usually struggle with accurately modelling out-of-sample and out-of-distribution uncertainty estimation. The proposed approach is to generate pseudo-inputs outside of the support of the training data and a loss that encourages the net to revert with its predictions on these "observations" to a prespecified prior. To accomplish this, they rely on a method introduced by Stirn & Knowles (2020), who model a latent model precision $\lambda$ via a Gamma distribution. Relying on this model, the authors introduce a _Pseudo-Input Generator_, which, based on a given data density generates OOD observations via gradient descent into low-density regions.


**Broader Impact Concerns:**

The broader impact statement is just a copy and paste of the template. While I agree that there are no broader concerns in the paper, the template should be removed in any final version.


**Requested Changes:**

The following two bullet points are critical before I can recommend acceptance
- Discuss the uncertainty limitation mentioned in (W1) more prominently to make it clearer to the reader that its primary focus is solely the OOD uncertainty improvement.
- Incorporate the comments raised in the other questions and weakness comments wrt the experimental setups and evaluations

Solving/answering the remaining questions and weaknesses mentioned above will further strenghten the paper in my opinion, but are not critical to acceptance.


### Typos and other minor things
- Sec 2.1 "predictor is indeed forced" -> it is encouraged by the D_KL term not forced which implies a hard constraint
- Fig 1: The legend is broken lacking a description of what the green/red lines represent.
- Abbreviations BBB, NCP, and IP are all only introduced indirectly
- Typo 1.2 2nd paragraph: Lastly, such **a** model's likelihood
- (very minor) Fig 4: More a computational graph rather than a graphical model under which I would instinctively assume a generative plate diagram.
- Sec 2.2 1st par: this intuition generalises to higher dimension**al** experiments.
- Alg 1: $\delta$ is not introduced and hidden in the appendix; p_data becomes p;
- Table 1: \tilde y is never introduced
- Table 2: log p(x) reported with too high precision in the decimals compared to the reported stds
- Several references are broken (not checked for completeness), e.g., Ian Goodfellow vs Ian J Goodfellow, P-A Mattei vs Pierre-Alexandre Mattei, Xu et al. is cited twice with the same paper,...


**Strengths And Weaknesses:**

### Strengths
- The paper tackles an important task in a well-motivated manner and seems[1] to improve upon its motivating predecessor VV (Stirn & Knowles, 2020)
- The proposed procedure for generating pseudo inputs can be run before the main training routine starts and the points can be generated in parallel for further efficiency.
- The paper is well written (and has very nice high-quality plots!)

### Weaknesses
- (W1) The title "robust uncertainty estimates...", abstract ("yields robust and interpretable predictions of uncertainty"), and several other statements in the paper suggest general improvements in the uncertainty estimation, i.e., a better calibration overall. However, as the authors acknowledge at the end of Sec 3.1, its in-distribution calibration is worse than prior work.
- (W2) The authors claim in their abstract "state-of-the-art performance on diverse tasks", which is not given in the main paper. The UCI regression improves only upon its VV predecessor and only in the OOD task. The generative MNIST/SVHN/CIFAR experiments are only compared against a plain VAE model, and even there suffer from decreased performance wrt log-likelihood in two of the three setups. The generative setup further lacks especially a comparison against the V3AE predecessor.
- (W3) As already mentioned in (W2), the experimental setup lacks a deeper comparison wrt more modern approaches. E.g., for the UCI regression setup and the authors' setup of comparing sampling-based and sampling-free methods there are several sampling-free BNN approaches (e.g., Wu et al., 2018; Haussmann et al., 2020), as well as more modern sampling-based BNNs than the plain BBB (e.g., Daxberger et al., 2021; ...). Similarly, the authors mention several OOD pseudo-estimation approaches in Sec 1.1, which fall in their direct uncertainty estimation category and would be interesting as comparisons that take a different approach than the baselines compared against in this category.
- (W4) The experiments are all using rather small neural nets.

____
Daxberger et al., Bayesian Deep Learning via Subnetwork Inference, ICML 2021
Haussmann et al., Sampling-Free Variational Inference of Bayesian Neural Networks by Variance Backpropagation, UAI 2020
Wu et al., Deterministic Variational Inference for Robust Bayesian Neural Networks, ICLR 2019


### Minor questions and comments
- Q: The authors claim a "holistic evaluation" in the abstract, what do they mean by that formulation?
- MC Dropout and BBB are introduced/labelled as ensemble methods and combined with the deep ensembles of Lakshminarayanan et al., which seems suboptimal in my sense. The approach of Lakshminarayanan et al. trains a true ensemble of models, and BBB trains a single model. It does require sampling, but that is simply due to the simple lack of an analytical solution to the respective expectations and not an ensemble in its classical sense, e.g., as in the "deep ensembles" paper.
- Q: What is the specific setup for the BBB in Figure 2 and elsewhere? E.g., it did not become clear to me whether it also has access to a heteroscedastic likelihood or not.
- Q: The authors mention that simple GMM seems sufficient to estimate p_data, have they explored more flexible, but also more expensive methods, e.g., flow-based approaches and are they also not beneficial to this rather straightforward approach? If the GMM is indeed already so stable, I would suggest making that statement more prominent as it shows that the constraint of having a p_data that might worry some readers is not such a huge constraint.
- Sec 1.2 paragraph on Ensembles. The authors mention several papers and Figure 2 in relation to that paragraph, but the Figure visualizes non of these methods.
- Q: How are a,b chosen in the prior and how sensitive is the model to their choice?
- Q: As mentioned in (W4), the experiments all rely on rather small networks. Can the authors comment on the scalability of the method, especially wrt the performance (the additional computational costs seem to be minor compared to BNNs if I see it correctly)? Given that larger models tend to become even more overconfident it would be interesting to see how the model copes with that and whether the approach can be even more beneficial in such settings.
- Q: The authors claim the ELBO as a commonly reported approximation to the marginal likelihood. Can the authors provide some references to that? I would claim that the default is still to approximate the marginal likelihood via sampling for comparison with other approaches.
- Q: Figure 5: What was the VI setting used in this approach? And how would it compare to e.g, the sampling-free approach by Wu et al., or the sampling-based by Daxberger et al., who have the benefit of dropping the mean-field constraint of a plain BBB approach?
- Q: Figure 5: How would VV perform in this example?
- Q: I am probably overlooking something about Figure 6, but can the authors comment on why the aleatoric estimate is so bad, given that it is a lot more varied in the raw data in Figure 5? (Or is that just a visualization artefact due to the scale of Fig 6?)
- Q: Fig 6 seems broken or a rather bad aleatoric estimate, which remains essentially constant while the epistemic should be lower in the in-distribution?
- Q: Can the authors comment on (W2) in the context of Table 1 and the statement in Sec 3.1 that the model "performs best in its class"?
- Q: Fig 7/8 lack a comparison against Skafte/Detlefsen et al. who already perform similar experiments with similar qualitative performance (at least judging from the figures)
_(It might be worth doing a double check wrt the Skafte et al. reference to make sure that the reference is correct, the author seems to be credited in the cited NeurIPS proceedings as "Nicki Skafte", while in the publication itself as "Nicki Skafte Detlefsen")_
- Q: Can the authors comment on the switch from ReLU to ELU in the UCI regression setup? The former would make it more comparable to the standard setup introduced by Hernandez-Lobato et al.. Similarly, even accepting the ELU, the resulting log likelihoods and RMSEs seem rather low and very different to prior work.
- Q: UCI Results in the appendix, e.g., Table 12,13: Why is BBB reported with nans in several data sets?
- There is a promise of supplementary material with code and prior parameters, neither of which is provided.
- Q: App III..5 mentions that d-VV and VV were run with a reduced number of trials due to the tightness of the deadline. Is this statement related to a previous submission as TMLR should not have this deadline constraint? And if so do the current results have the same number of trials as the baselines?
This statement also seems to indicate that the individual methods were run on different train/test splits in the UCI experiments. Given the sometimes rather large performance difference between different train/test splits into some UCI data sets it would be a lot cleaner to compare all experiments on the same train/test splits for each data set.

## Summary
The paper is overall well written and has an interesting idea that can be applied rather directly. However, a major downside of the paper is its experimental evaluation demonstrating the usefulness/necessity of the proposed idea. Once such a detailed evaluation has been performed the paper will provide a nice addition to the literature. In its current status, it has not reached that final form yet, but the authors are highly encouraged to evaluate these missing pieces to arrive at a well-rounded paper.

---

### Review · Reviewer_gYKg · 2022-08-25

**Summary Of Contributions:**

The paper studies uncertainty estimation in machine learning. Traditional ML approaches often extrapolate arbitrarily in the OOD regime, resulting in unreliable uncertainty estimation. To address this issue, the authors propose a new method to obtain reliable uncertainty estimation for out-of-distribution (OOD) samples. The proposed method has two key components: 1. Generating Pseudo-Inputs in the low-data density regime via a gradient-based method (Algorithm 1) 2. Introducing the so-called disperse loss that encourages high-entropy in the OOD regime ("know what they don't know", see Eq (4).) Finally, the authors also demonstrate the strength of their methods on a variety of tasks, e.g., toy task, UCI, and generative models.

Overall, I find the paper relatively well-written and make some interesting and insightful contributions to "ML + Uncertainty Estimation."

**Requested Changes:**



- For many figures in the paper, more explanation will be appreciated. E.g., what do green/red curves represent in Fig.1? What do the legends BBB, and NCP mean in Fig2? I couldn't find the definition in the paper.

- "The predictive performance of machine learning models has drastically increased in the past decade, but
the quality of the accompanying uncertainties have not followed." Can the authors provide a reference for the latter statement? A line of work empirically shows that better models (in particular, scaling up a model size) have better uncertainty estimates.

- The organization of sec 1.1 (background and related work) and sec 1.2 ( Robust uncertainty estimates) seems confusing. Sec. 1.2 also belongs to the "background and related work" category, right? In addition, the introduction seems a bit too long (3.5 pages while the main text has only 10 pages)

-  After equation (1):"Its variance V(x) ...", the sentence reads like "the decomposition of the variance into the product of epistemic and aleatoric" is a well-defined theoretical result. However, from the referenced paper, the interpretation of this decomposition is only a heuristic. Please clarify.

- Sec 3.1.1. Can you provide more details regarding the " distributional shift" in the toy experiments?


**Strengths And Weaknesses:**

## Strength

- The proposed methods seem to be simple, interpretable and principled, although no theoretical insights were made to justify it.
- The experimental results (Fig 3, 5 and Table 1) seem to provide convince evidence supporting their method.

## Weaknesses

- In order to generate pseudo-inputs, we need to have access to the data distribution $p_{data}$, which is often unrealistic in practice. E.g., how can I get OOD uncertainty estimation for image classification (ImageNet) using the current method? Note that the authors discussed this limitation in the conclusion section of the paper. However, I would recommend highlighting this limitation in the introduction as well.

- The idea of using pseudo-inputs to generate uncertainty estimation is certainly not new. Is there an experimental setting that we could fairly compare such methods against the proposed method?

---

### Review · Reviewer_zzMd · 2022-08-29

**Summary Of Contributions:**

This paper addresses the issue of providing reliable uncertainty estimates out of distribution with neural network models for regression tasks. The proposed method builds upon the method of Variational variance (VV) of Stirn & Knowles (2020). VV consists of using a heteroscedastic Gaussian likelihood while placing a factorised Gamma prior on each observation’s noise variance. The posterior over this Gaussian likelihood variance is then estimated with a variational amortisation network. The present paper notes that the amortisation network can extrapolate poorly out of distribution. To correct this, the authors introduce a series of out-of-distribution pseudo-inputs at which the amortisation network that predicts the variance posterior is encouraged to place mass on large variance values. To choose pseudo-inputs, the authors propose to fit a density model to the covariates, sample from the model, and take steps in the direction of the negative log-density to push the particles towards lower density regions. A GMM is used as the density model.  The authors compare their method against VV and alternative Bayesian deep learning procedures on supervised UCI regression tasks and unsupervised generative modelling with VAEs. In both cases, the proposed method is used as a drop-in replacement for the standard homoscedastic Gaussian likelihood function.

*Andrew Stirn and David A Knowles. Variational variance: Simple and reliable predictive variance parameter- ization. arXiv preprint arXiv:2006.04910, 2020.*


**Requested Changes:**

**Clarity of exposition**

* Clarify notation fixing the issues described above: distinguish clearly between inputs and targets, distinguish between single observations and full datasets, expand objective function so it is clear what you are computing, add missing notation in algorithms, etc

* I would suggest dropping the pseudo Bayesian interpretation of the proposed method and explaining it more directly as a form of regularisation based on enforcing large errorbars out of distribution.

* Expand a discussion on aleatoric and epistemic uncertainty to clearly explain that the proposed approach models epistemic uncertainty in the noise variance but not in the predictive mean. This is a key detail that distinguishes the method from most comparable other approaches but is glossed over.

* Add details to plots. Figure 1 is presented at a point in the paper where no methods have been introduced and thus it is difficult to take anything away from it. Additionally, the red and green lines are not defined in the legend. Figure 2 is an illustration of a pitfall of the NCP method but this method is not explained anywhere. In figure 3, the idea of prior and data uncertainty mentioned in the legend is not explained clearly enough. In Figure 5 there are two sets of green and red lines of different tones. What do these refer to? Also, the legend uses very similar colours for prior and ground truth making these difficult to distinguish from each other.  What are the continuous and dashed lines in figure 6?

* Add experimental details. The maintext currently has very little details on the experimental setup. How are hyperparameters selected? What train val test splits are used for UCI? How are the OOD rejection plots in Figure 10 computed?

* Do not compare methods’ performance in terms of training loss (ELBO) and training KL. Different methods use different losses and even if they used the same one this would just measure thoroughness of optimisation.

**Additional experiments**

* Add a comparison to other methods that use pseudo-inputs to check which pseudo-input generation method works better.

* Move the evaluation of the effect of number of pseudo-input optimisation steps on performance from Appendix table 4 to the main text. Discuss why more steps seem to perform better in that experiment, seemingly contradicting the assumption that pseudo-inputs need to be close to high data density regions.
* Add an experiment evaluating the effect of the number of pseudo inputs on performance
* Add an experiment evaluating the effect of increased input-space dimensionality on performance for fixed number of pseudo-inputs.

* Add VV baseline for VAE experiments.

**Strengths And Weaknesses:**

**Summary**

This paper presents some interesting ideas but has major drawbacks in terms of exposition and experimental evaluation. The paper clearly needs more work before it is suitable for publication.

**Strengths**

* This paper expands on a growing literature which aims to assign large uncertainty to out of distribution points by training on a set of pseudo-inputs placed outside of the high density region of input space. However, differently from existing work, which uses somewhat naive approaches for selection of pseudo inputs like adding noise to the train data, this paper guides pseudo-input generation with an explicit density model. This seems like a good idea to me.

* This paper argues that NNs can extrapolate from a few out-of-distribution pseudo-inputs and thus learn to assign high-uncertainty out of distribution. This would allow avoiding having to cover the whole of input space with pseudo-inputs. If the supposition were true, we could enforce large uncertainty at a set of pseudo inputs at the boundary of the training data density and expect reasonable uncertainty out of distribution. This argument seems sound to me, since similar observations have been made in previous work (Lee et. al. 2018). However, as described below, I think additional experiments checking this hypothesis are required.

* Apart from standard regression tasks, the authors demonstrate the utility of their approach for constructing a VAE likelihood function, showing the generality of the approach.

**Weaknesses**


1. Clarity and soundness of presentation
	* Clarity of writing
		* The notation used in the paper is ambiguous in many places. In general, the task being solved is a problem of regression of covariates x onto targets y. However, all of the notation is written only in terms of x. No where is x defined to be the set {x, y} — although this is what I think the authors intend — but even if it were, there would be ambiguity due to the x being contained in the set {x, y}. This leads to some strange expressions like in section 1.2 $p(x) = \mathcal{N}(x; \mu(x), \sigma(x))$; how can a distribution over a variable be conditional on that same variable? There is also some overloading between the probability of observing a specific value of $x$ and a whole dataset in equation 2. Also, In algorithm 1, the step size $\delta$ and the ”tolerance” variable are undefined.

		* The expressions for the quantities being computed are omitted from the main text. Losses 4 and 5 are written only in terms of probability distributions  p and q without specifying what distributions these are or how they are parametrised. This makes it hard for the reader to get an idea of what the authors are computing exactly or how to re-implement the method.

		* Almost all of the plots in the paper lack sufficient explanation for the reader to be able to gain much from them. I go into detail in the requested changes section.


	* Soundness of method formulation and explanations: this method builds upon VV of Stirn & Knowles (2020).  Note that that paper is also currently under review.  If I understand correctly, this method places an observation-wise prior over the likelihood function’s variance ($\lambda_n$ with $n$ denoting observation index). However, this prior factorises across observations. (Sidenote: this is not clear in the notation used since $p(\lambda)$ and $p(\lambda_n)$ are treated interchangeably.) A key issue of the proposed method which is inherited from  Stirn & Knowles (2020) is that they assume a factorised prior across input space $p(\lambda_1, \lambda_2) = p(\lambda_1) p(\lambda_2)$. This corresponds to an assumption that observing a target at one input does not tell us about the likelihood variance at any other input. Thus, the posterior is trivial; it becomes the prior at all non-observed points. The proposed method works in practise because the amortisation network does not have enough flexibility to capture the true posterior described.  Probabilistic models should be sensible on their own and should not require some artefact of approximate inference in order to work well.
Thus, while I think that the proposed method is sensible on its own — generating pseudo-inputs using a density model and assuming the uncertainty predicting NN can extrapolate reasonably — I do not believe the probabilistic model presented to  be sensible. I think that the current exposition can seriously confuse readers.

3. Alignment between claims and experimental validation
	* A key motivation for this paper is the unsuitability of alternative pseudo-input selection methods. For instance, the shortcomings of noise contrastive priors are illustrated in Figure 2. However, no empirical comparison is made with any other methods of this sort.

	* This paper relies on the key assumption that NNs will extrapolate well from a small number of out of distribution pseudo-inputs placed near the training data; meaning training the NN to output a large uncertainty at these points will be enough for it to output a large variance parameter at all out of distribution points. As far as I can tell, this key assumption is not tested anywhere in the paper except for the toy 1d experiment of figure 3. I think it is important to have an evaluation of the performance of the method in terms of the distance of the pseudo inputs to the train data and in terms of the dimension of the input space, since we would expect a pseudo-input method to suffer from the curse of dimensionality in high-dimensional input spaces.

	* The authors employ the ELBO as one of their evaluation metrics. This is not sensible for 2 reasons; 1) the ELBO can be an arbitrarily bad proxy for the model evidence (this is also stated by the authors)  2) the probabilistic model is not very sensible and thus the model evidence is somewhat meaningless.

	* The authors of the VV paper also apply their method to VAEs. Since the proposed method is an extension of VV, it is not clear to me why VV is does not appear as a benchmark in the VAE experiments.

	* On UCI regression experiments, the authors claim “Our method performs best in its class,” but from table 1 it looks like it doesn’t except on metrics that only involve the training data. I am quite confused by this.

---

### Author Response · Authors · 2022-09-08
**Upcoming exchaustive answer to the reviewers' comments**

We'd like to thank the reviewers for their thorough and constructive reviews.

We have not yet come up with an extensive answer to the questions raised, and come up with proposals to address the identified weaknesses. This was due with the timing of the received reviews colliding with our holidays schedule.

It is unfortunate that the final decision timeframe opens up so swiftly after reviews have been gathered.

Through this message, we want to give notification to the reviewers that we will be posting an exhaustive answer to their feedback on the 13th of September.

Thank you for your understanding,

Best regards

---

### Author Response · Authors · 2022-09-13
**Answer to reviewer gYKg [1/2]**

We’d like to thank reviewer gYKg for his review. The identified weaknesses and requested changes are all relevant, and demonstrate a good understanding of our proposed contribution.

Our answer will first address the requested changes, providing direct clarifications, details and answers where possible through the reviewing system. For more general comments and changes, we will only provide succinctly the essence of the changes we envision to satisfy the reviewer, as we want to answer all comments as swiftly as possible. If the reviewers indicate that such changes would lead to a potential acceptance of the proposal, we would be able to submit a revision by the 26th of September, containing suggested changes to the body of the document.

—

* __On the explanation of figures__

We acknowledge the shared request between all reviewers for more explanations related to the provided figures. The amount of provided explanations was limited to what we believed to be the most succinct while remaining understandable. This was motivated by a desire to keep the body of the proposal to a maximum of 10 pages. The TLMR format allows for more flexibility, which means that we can expand on the figures whose descriptions were identified as too concise. This includes figures 1 (gYKg, zzMd), 2 (gYKg, obnL, zzMd), 3 (zzMd), 5 (obnL, zzMd), 6 (obnL, zzMd).

Generally, we can add for each figure a few sentences guiding the interpretation of the Figure.

For example, for Figure 1, after correcting the legend, we could add such a sentence:
As seen in Fig. 1, the regularisation of the model with the KL divergence minimisation on pseudo-inputs (green line) enforces a consistent extrapolation, contrary to a conventional model (red lines) which would extrapolate arbitrarily depending on training random conditions.

* __On providing reference to support "The predictive performance of machine learning models has drastically increased in the past decade, but the quality of the accompanying uncertainties have not followed."__

The point we wanted to make here can indeed be misunderstood due to the formulation of that sentence.

We propose to rephrase it as

The predictive performance of machine learning models has drastically increased in the past decade, but the quality of the accompanying uncertainties have not followed __at the same pace__.

Which would put more emphasis on the fact that the speed of the improvement of the uncertainty estimates has not matched the speed of improvement of the general predictive power of models.
Several references included in the introduction of Sec. 1.1 provide supporting arguments. For example Eric Nalisnick, et al. Detecting out-of-distribution inputs to deep generative models using typicality, 2019., shows that recent models are displaying wrongful higher likelihoods on out-of-distribution inputs.

* __On the organisation of section 1.1 and 1.2 and balance between size of introduction and main body__

Section 1.2 has been separated from background and related work, as it provides the base notations and concepts on which our method expands. It does detail some related work, but which is essential to the formulation of our proposition. We could easily make that distinction clearer by adding a simple sentence that would explain the scope of that section.
We further believe that the addition of details about the figures and formulations as requested by all reviewers will expand the content of the main body, and balance its respective size against the introduction.

* __On the equation (1) and decomposition of uncertainty__

This nuance was correctly understood by the reviewer. We can clarify the concept by modifying the sentence that follows equation (1). We propose the following:

Its variance Var[x] = (β/α)·(α/(α − 1)) can explicitly decomposed into an aleatoric β/α and an epistemic term α/(α − 1) (Jørgensen, 2020, p16), which would offer a direct verification of whether a model knows what it knows. __This decomposition stems from the convergence of α/(α − 1) towards a constant 1 as the number of observations grows towards infinity and seems to be empirically confirmed on Fig. 6.__

* __On adding more details regarding the distributional shift in section 3.1.1__

A distributional shift appears whenever the training dataset and testing dataset were not generated from a similar distribution. In the case of section 3.1.1 the distributional shift arises from the introduction of a hole in the training dataset, while maintaining a testing dataset uniformly distributed.

We propose to introduce the following sentence for clarification:

A distributional shift is herein introduced in the form of a hole in the otherwise uniformly distributed training distribution. This is indicated on the bottom row of Fig. 5 by the absence of any training data point between the dashed lines.

---

---

> ### Author Response · Authors · 2022-09-13
> **Answer to reviewer gYKg [2/2]**
>
> We would also like to quickly address some of the raised weaknesses in the hope of showing that we are grateful for the constructive feedback that was received.
>
> * __On the need to have a good distribution for generating pseudo-inputs__
>
> In the use cases we explored, our findings suggest that the generation of useful pseudo-inputs does not require the usage of a perfect of a distribution that matches with the training data. As long as the distribution is able to outline the boundaries of regions of high density for the data, it will enable the usage of the proposed method. We further don’t see a scalability issue with that approach, as it reduces down to the generation of adversarial inputs for e.g images, which have proven to be interpreted by models as being outside the training support.
>
> * __On the idea that pseudo-input generation is not new and requires comparison against other previous approaches__
>
> Multiple reviewers (zzMd, obnL, gYKg) have raised this point. Wherever mentioned in other reviews, I will refer to the following answer. We do provide in Tab. 6 in annex, a comparison of our pseudo-input generator with the most common method otherwise used, which consists in adding Gaussian noise to the existing inputs. For more complex methods, such as the one introduced in Lee et al. (2017) it is unfortunately impossible to evaluate the influence of the pseudo-input generator alone as it is tightly coupled with the rest of their method. For example, in the case of Lee et al., the generative model that generates pseudo-inputs is jointly trained with the classifier, making the comparison of the pseudo-input generators impossible.
> It would be interesting to come-up with a metric for evaluating the performance of a given pseudo-input generator as is, without having to rely on a downstream overall performance. We could unfortunately not come up with a definite way of measuring pseudo-input generation performance.
>
> We suggest adding some clarification to section 2.2 to make this nuance clearer.

---

### Author Response · Authors · 2022-09-13
**Answer to reviewer obnL [1/3]**

We would like to thank reviewer obnL for the depth of his review and his attention to details.
All reported “typos and minor things” have been duly noted. They are all relevant and will thus be integrated in our future revision.

The requested changes mostly appear as answers to the multitude of raised questions. We will thus start by addressing the first requesting change and answer all the questions raised. Due to the number of questions, we will not be able to provide as much detail as was provided for the answer to reviewer gYKg, but are happy to do so if reviewer obnL requires more details on specific points.


* __On making it clearer to the reader that the main focus is the improvement of the OOD uncertainty estimation__

We agree with the reviewer that our proposal would benefit from focusing our claims on the improvement of the uncertainty predictions out-of-distribution. In practice, that would amount to adding some nuance to the sentences that can be perceived as over-claiming the contribution of our proposal. As correctly identified, the abstract is the first sentence we would update. It should become instead:

With a holistic evaluation, we demonstrate that this yields robust and interpretable predictions of uncertainty, __with intuitive out-of-distribution extrapolation properties while preserving predictive performance__ on diverse tasks such as regression and generative modelling.

---

> ### Author Response · Authors · 2022-09-13
> **Answer to reviewer obnL [2/3]**
>
> __Answer to all questions:__
>
> * On the meaning of holistic evaluation:
>
> By holistic evaluation, we mean that the proposed evaluation of the main regression experiments is based on a multitude of metrics, that each offer an evaluation of the compared models from different aspects. To the best of our knowledge, in the field of uncertainty quantification, there is no agreed upon baseline nor unique metric that uniquely defines the performance of a given model. We thus decided to integrate all metrics seen in literature to compare the models on a variety of criteria.
>
> We can make this choice more obvious by rephrasing part of the introduction to section 3.
>
> As in Stirn & Knowles (2020), we propose to assess it using multiple metrics. The use of multiple metrics, as identified in the different baselines we used, allows for a holistic comparison of our method against previous models. It also reflects the absence of a definite baseline or single metric for the evaluation of out-of-distribution uncertainty quantification.
>
> * Are BNNs ensembles?
>
> We decided to group BNNs with ensemble models as they do provide, in theory, a continuous ensemble of models from which a single prediction is extracted. In practice, the resemblance is even more striking as they require sampling, and the extracted prediction is indeed ensembled from different models.
>
> * On the BBB setup for Figure 2.
>
> See the first point of our answer to reviewer gYKg for adding details to the figure interpretations.
> In this case, we simply apply the original formulation of BBB, whose variance is expressed as a function of the mean prediction, as in any ensemble model. In that sense, yes the likelihood is heteroscedastic. On Fig. 2, is only represented the mean prediction, as the point that we’re trying to make is that this interdependence is detrimental to the ability to accurately estimate uncertainty. If the uncertainty estimates are improved, which would happen when we increase the regularisation level, then the mean prediction is deteriorated.
>
> * On the exploration of other methods for estimating p_data
>
> We have explored the usage of flow-based likelihood methods. This exploration was realised in an explorative phase and the results are not directly fit for addition to our contribution. Our findings suggested that the added complexity was not rewarded in terms of performance.
>
> * On the choice of prior parameters
>
> We refer to the Appendix III.8 and IV.5 for the experimental setup used. The choice of prior parameters is by definition use case dependent, and simple heuristics were used here.
>
> * On the size of the networks
>
> Our method is not bound theoretically by computational considerations.
> By controlling p_data prior to training and through its independence with the different training runs, we can easily interchange simple neural networks with larger ones for the regression or generative task. We also show in the annex in Tab. 20 & 21 that the pseudo-input generator increases in that instance the training time by 26%.
> The main point of our paper being that the training of the model’s uncertainty on pseudo-inputs result in better out-of-distribution training, we do not judge necessary to demonstrate that this theoretical scalability is verified empirically.
>
> * On the usage of the ELBO as an approximation for the marginal likelihood
>
> We mostly agree with the underlying criticism that the ELBO is not the best approximation that exists for the marginal likelihood, and acknowledge it in the body of our proposal (Introduction of Sec. 3). As hinted above, we ended up using it in the set of metrics used for evaluation, as it has been used in the past to compare model performances. For that, see the original VAE publication Diederik P Kingma and Max Welling. Auto-encoding variational bayes, 2013 or Lars Maaløe et al. Biva: A very deep hierarchy of latent variables for generative modeling, NEURIPS 2019
>
> * On the VI setup for Figure 5.
>
> For the closed form of the ELBO in a regression setting, see Sec. III.1 of the annex.
> The prior was chosen to be homoscedastic, corresponding to an uncertainty level displayed by the grey shade in Fig. 5. We could add a subsection in the Annex that details a bit further the setup as done for the UCI benchmarks.
> Regarding the comparison with the others, we cannot at this moment answer, as we have not had time to read through these additional proposals.

---

> > ### Comment · Reviewer_obnL · 2022-09-14
> > **Answer to: Answer to reviewer obnL [2/3]**
> >
> > Thank you for your detailed answers, I will only comment on a subset of them.
> >
> > > By holistic evaluation, we mean that the proposed evaluation of the main regression experiments is based on a multitude of metric
> >
> > My complaint here was mostly meant towards the term _"holistic"_. This term implies a universality. As you state, there is no fully agreed metric. You use a several metrics/a diverse set of metric/... for your evaluation. But this is not _a holistic evaluation_, whatever that term would even means in a machine learning context apart from a proof.
> >
> > > We decided to group BNNs with ensemble models as they do provide, in theory, a continuous ensemble of models from which a single prediction is extracted. In practice, the resemblance is even more striking as they require sampling, and the extracted prediction is indeed ensembled from different models.
> >
> > Fair enough, but then you can refer to any type of marginalization always as an ensemble which is not the common way people use ensemble in the literature. (But this is a rather minor point)
> >
> > > In this case, we simply apply the original formulation of BBB, whose variance is expressed as a function of the mean prediction, as in any ensemble model. In that sense, yes the likelihood is heteroscedastic.
> >
> > I am not sure I fully understand the provided explanation which reads to me as if you refer to something like the variance in the uncertainty around the mean, e.g., $\text{var}(\mu(x))$? If so this is not what the term means. Blundell et al. (2015) do not specify their likelihood explicitly, which is why I would assume the common usage of a homoscedastic likelihood. For a model with additive Gaussian noise we have a homoscedastic likelihood as $N(y|\mu(x),\sigma^2)$ and a heteroscedastic one as $N(y|\mu(x),\sigma(x)^2)$. Does the model follow the former, or the latter?
> >
> > > We refer to the Appendix III.8 and IV.5 for the experimental setup used.
> >
> > III.8 does not refer to the $\Gamma(\lambda|a,b)$ used throughout the paper, but only to $\alpha, \beta$, Similarly IV.5 does not give any details on the grid that was used.
> >
> > > On the size of the networks
> >
> > This question was not related to the computational cost, but the predictive performance and the problem of overconfidence larger models can develop. (See the calibration paper on this topic by Guo et al. (2017))
> >
> > > We mostly agree with the underlying criticism that the ELBO is not the best approximation that exists for the marginal likelihood
> >
> > Using a metric you agree is suboptimal just because other have done it seems somewhat suboptimal as an argument ;)
> > Especially if it is the main metric your model improves upon the baseline approaches in Table 1.

---

> ### Author Response · Authors · 2022-09-13
> **Answer to reviewer obnL [3/3]**
>
> * On the performance of VV on Figure 5.
>
> VV is represented on Figure 5 by the red results. As explained earlier, we will add explanatory comments related to the different figures.
> Briefly put, the issue with VV highlighted by Figure 5 is that the lack of OOD regularisation means that it does not rely on prior information for extrapolation. This is shown by the fact that the uncertainty outside the training range does not increase towards the grey prior level. That is particularly problematic when we introduce the distributional shift (bottom row). The VV’s uncertainty does not account for this shift.
>
> * On the aleatoric performance on Figure 6.
>
> We are not sure to understand the confusion of the reviewer here. Again, we believe that the intended addition of figure explanations would resolve it.
> On Figure 6, the aleatoric uncertainty, represented in yellow, needs to be factorised with the epistemic uncertainty represented in dark to obtain the overall uncertainty as displayed on the top row of the right section of Figure 5. In that right section of Figure 5, each coloured line represents the uncertainty of the one of the models displayed on the two rows on the left.
> We deem that the aleatoric uncertainty behaves as expected as it grows steadily within the training support.
>
> * Comparison against Skafte et al. in Fig. 7 and 8
>
> It will not be possible for us to easily include the work of Skafte et al. in Fig. 8 as this experiment was not part of their contribution, and their code is therefore not suited to that experiment.
> For Fig. 7 and the general VAE experiment, we became aware that some reported results are unrealistic, and chose not to include the method in.
>
> * On the usage of ELU vs ReLU
>
> This difference is explained by our choice to follow the experimental setup of our baselines.
>
> * On the presence of NaNs for BBB
>
> The NaNs appear for metrics that could not readily be measured using the code of the BBB authors.
>
> * On the submission of code and supplementary materials
>
> We did forget to include the code in the original submission. A new revision has been published with the code.
>
> * On the mention of tightness of the deadline
>
> Thank you for picking this up. We mistakenly left this in our submission as an artefact of a previous submission.

---

> > ### Comment · Reviewer_obnL · 2022-09-14
> > **Answer to: Answer to reviewer obnL [3/3]**
> >
> > > It will not be possible for us to easily include the work of Skafte et al. in Fig. 8 as this experiment was not part of their contribution, and their code is therefore not suited to that experiment. For Fig. 7 and the general VAE experiment, we became aware that some reported results are unrealistic, and chose not to include the method in.
> >
> > Sorry, that was my formulation error. I originally meant your Fig 7 and their Fig 8. However, the second statement is interesting. Can you further elaborate on that? Or, independent of whether you want to or not, it would be cleaner to report (main text or appendix) that you were not able to reproduce the results the authors' claim to make it clear that a missing comparison is not an oversight, but an actual choice.
> >
> > > On the usage of ELU vs ReLU
> > > This difference is explained by our choice to follow the experimental setup of our baselines.
> >
> > Fair enough, but the main text reads as if the experimental setup follows Hernandez-Lobato et al. and Skafte et al., neither of which use ELUs. With following the baseline you most likely refer to Stirn and Knowles, who again only mention this switch in the appendix, but gave a specific argument for it. Similarly your answer does not discuss the discrepancy with published results, e.g., the ones provided by Stirn and Knowles.
> >
> > > The NaNs appear for metrics that could not readily be measured using the code of the BBB authors.
> >
> > Reimplementing BBB should be a rather trivial task. Even if not adapting many of the existing implementations (I have not found the one provided by the original authors) to get the desired metrics should not be a problem.

---

> ### Comment · Reviewer_obnL · 2022-09-14
> **Answer to: Answer to reviewer obnL [1/3]**
>
> >  but are happy to do so if reviewer obnL requires more details on specific points
>
> I would be happy if the authors could comment on the main point, i.e., the weakness in the experimental evaluation. How can the claimed state-of-the-art performance be justified in the light of the points mentioned in (W2), (W3) and similar remarks by the other reviewers?

---

> > ### Author Response · Authors · 2022-09-22
> > **Thanks for the clarifications**
> >
> > Thanks a lot for taking the time to provide clarifications on your comments. We'll answer the points that seem the most relevant.
> >
> > > My complaint here was mostly meant towards the term "holistic"
> >
> > Thanks for the clarification. We could indeed replace that term with something more along the lines of "complete set of metrics", which would make it read less as if we were claiming that our evaluation is universal.
> >
> > > I am not sure I fully understand the provided explanation which reads to me as if you refer to something like the variance in the uncertainty around the mean ...
> >
> > Right, sorry, we misunderstood what you meant here. In this case, the original BBB would have access to an heteroscedastic likelihood. See sggm/baselines/nn.py in the added code.
> >
> > > III.8 does not refer to the $\Gamma (\lambda\vert\alpha,\beta)$ used throughout the paper ...
> >
> > Our notation indeed got confused in that appendix, $\alpha$ can be interchanged with a, and same for $\beta$ and b.
> >
> > > it would be cleaner to report (main text or appendix) that you were not able to reproduce the results the authors' claim to make it clear that a missing comparison is not an oversight, but an actual choice.
> >
> > This is a great suggestion, and we shall include it in the revision.
> >
> > > I would be happy if the authors could comment on the main point
> >
> > At this point we will not be able to radically change the experimental setup. We therefore suggest rephrasing that claim, and putting more emphasis on the improvement of the pseudo-input generation procedure, which shows as giving the possibility to better control the extrapolation of the model without loosing too much of the predictive power.

---

### Author Response · Authors · 2022-09-13
**Answer to reviewer zzMd [1/2]**

We would like to thank reviewer zzMd for the relevant review.

The structure of the review makes it easy for us to reply. The most important identified weaknesses echo requested changes. We will therefore directly provide elements to answer the requested changes.

As explained in the answer to reviewer gYKg, we answer here all concerns raised, and if an agreement is reached with the reviewers, we will submit a revision before the 26th of September. That is the end date for the period during which reviewers can submit their recommendation.
For larger re-writes, such as the addition of new experiments, we cannot guarantee that we will have time to deliver them before the 26th of September.


* __On the clarification of mathematical notations__

Indeed, the adopted notation, as described in Sec. 1.2 under the title notation, we adopt x = {x, y} for the notation. What changes between the two notations is the font style. We thought that the notation section would make it clear to the reader that there is a difference between the different types of x. We suggest changing the notation to X = {x, y} to make that difference even starker.

The omission of the variable defined wrt Alg. 1, will be rectified.

Losses 4 and 5 are presented in their most generic form as this section explains the theoretical model underlying our proposal. As we are not limited by space constraints, we will add the expanded form of the losses that are currently provided in Annex in the adequate sections (Regression experiments and Generative experiments), as is commonly done in the literature.

* __On dropping the pseudo Bayesian interpretation__

We unfortunately have to push back on this suggestion from reviewer zzMd. Our proposal is based on a regularisation of a variational inference framework. With that regard, the theoretical grounding for our proposition would be difficult to understand without that framework.
We do nevertheless agree that the suggested interpretation has value, and we suggest adding a clarifying post-hoc sentence in Sec. 2.1 that would be as follows:

… hence the name of dissipative. __In a non-Bayesian understanding, the method can more simply be understood as adding a regularisation term that enforces large error bars out-of-distribution.__ The reliance of the model’s predictive …

* __On expanding the discussion around aleatoric and epistemic uncertainty.__

Thank you for the valuable suggestion. Indeed, we should add a point that underlines the specificity of the method with that regard. This would be combined with the addition of clarification regarding the decomposition of the uncertainty in its aleatoric and epistemic components as requested by reviewer gYKg.

It could take the form of:

… offers a direct verification of whether a model knows what it knows. __This decomposition also highlights one of the strengths of this framework. The mean prediction is by definition completely independent of the components that parametrise the predictive uncertainty. By opposition to ensemble models, this offers the potential to not have to deteriorate the mean predictive power of the model in order to reach better uncertainty predictions.__

* __On adding details to plots.__

See answer to reviewer gYKg

* __On adding experimental setups__

We recognise that the experimental setups can be expanded upon. Some elements requested are provided in the appendix, but in general, we shall include a paragraph in each section to detail the experimental setup used.

* __On comparing the models’ performance with the training ELBO and KL__

There might be a misunderstanding here. We do not report the training ELBO and KL in our metrics. These metrics are instead computed on a held-out test dataset. In the light of this confusion, we will add a simple sentence in the introduction of Sec. 3 to explain that.

---

> ### Author Response · Authors · 2022-09-13
> **Answer to reviewer gYKg [2/2]**
>
> * __On adding a comparison with other pseudo-input generation methods__
>
> We are unsure about what the reviewer means here by “works better”. Tab. 3 in annex already provides some elements for pseudo-input generation methods that lie within the same class as the one we used. See also answer to "On the idea that pseudo-input generation is not new and requires comparison against other previous approaches" to reviewer gYKg.
>
>
> * __On moving the evaluation of the effect of the number of steps in the pseudo-input generator to the main body__
>
> Without space constraints, we can very well follow that suggestion.
> Regarding the observation that the likelihood increases as the number of steps, a possible explanation is as follows. When pseudo-inputs are moved far away from the boundary with a large number of steps, the regularisation constraints are loosened. This means that the in-distribution uncertainty can match freely the uncertainty observed on training samples, and thus the likelihood of estimates on a held-out test dataset increases.
>
> * __On adding an experiment to study the influence of the number of pseudo-inputs__
>
> As detailed in section II.4 of the annex, the practical implementation is insensitive to the number of pseudo-inputs as each term is taken as the expected value over that set of points. It means that equal weight is provided to each. We therefore estimate that this added experiment would have very limited added value for our proposal. The addition of the expanded forms of the loss function used as mentioned above will make this nuance clearer, and we can very well also move the part in annex to the main body.
>
> * __On adding an experiment for evaluating the influence of the input-space dimensionality for a fixed number of pseudo-inputs__
>
> The influence of the input-space dimensionality on the pseudo-input generator can be seen in section II.3 of the annex. We recognise that it might not be clear that the different UCI datasets have very different input-space dimensions.
> We will adapt the table to make the dimensions of each dataset clearer.
> We will unfortunately not be able to add an extra experiment where a similar data generating distribution is used to populate training and testing datasets for different dimensions before the end of the review period.
>
> * __On adding VV to the VAE experiments__
>
> This is a good suggestion, and as hinted above, if the reviewers are showing signs that acceptance would be possible, we will be able to add VV to the VAE experiments before the end of the review period.

---

> > ### Comment · Reviewer_zzMd · 2022-09-16
> > **Thanks for the response**
> >
> > Thanks for the response. Unfortunately, I feel like some of my key concerns have not been addressed. I will summarise them here to facilitate discussion.
> >
> > * Lack of technical correctness: The proposed method is motivated by the formulation of a probabilistic model in which the prior factorises over input indices and the observations are assumed iid. For any model of this type, the posterior also factorises over input indices and thus the model is non-learnable. This situation is analogous to learning in a whitenoise process; observing the process evaluation at one timepoint does not tell you anything about the other timepoints. Unfortunately, this flaw is also shared by the unpublished work that the present paper builds upon (Andrew Stirn and David A Knowles. Variational variance: Simple and reliable predictive variance parameter- ization. arXiv preprint arXiv:2006.04910, 2020).
> >     * As a sidenote, the ELBO is only defined on the training data. Evaluating it on test data provides a somewhat difficult to interpret metic.
> >
> > * Lack of experimental substantiation for claims.
> >     * The authors motivate their method as improving over existing pseudo-input based methods but they don't compare their approach to any of these methods.
> >     * The authors make a series of key assumptions, like that their NN's will extrapolate well from training on a small number of OOD pseudo-inputs. These are never tested empirically outside of 1d toy settings.

---

> > > ### Author Response · Authors · 2022-09-22
> > > **Thanks for summarising your concerns**
> > >
> > > Thanks for engaging in a constructive discussion.
> > >
> > > * We are not sure to understand why the formulation of the probabilistic model discredits the proposition. Many models rest on assumptions that are violated with real-data while performing well-enough.
> > >
> > > * We agree that the ELBO is a somewhat imperfect metric on a held-out test dataset. We included it as it is used in the literature to compare models. As mentioned in the answer to reviewer obnL see the original VAE publication Diederik P Kingma and Max Welling. Auto-encoding variational bayes, 2013 or Lars Maaløe et al. Biva: A very deep hierarchy of latent variables for generative modeling, NEURIPS 2019.
> > >
> > > * Our claim for improving pseudo-input generation methods is based on the comparison we make with pseudo-input generated with Gaussian noise, which is the base of what the Contrastive Prior method does. For other methods, the comparison is impossible, as they're bound to the rest of the model.
> > >
> > > * We do include a measurement of the extrapolation beyond 1D within the UCI benchmark. Each experiment is doubled, once with the original dataset, and once with the dataset in which we deliberately introduced distributional shifts. For clarity, we regrouped both under one table, but we could also split it up.

---

### Decision · Action_Editors · 2022-10-06

**Recommendation:** Reject

**Comment:**

The reviewers found several strength in this paper, but argued it cannot be accepted in this form. They found that the paper needs to better substantiate the following claims:
- Improvements over existing pseudo-input based methods
- The assumption that NNs extrapolate from few points needs stronger evidence.

**Audience:**

This paper is of interest to the TMLR community.

**Claims And Evidence:**

This paper tackles the problem of uncertainty estimation with neural models in regression tasks. The method is evaluated on a range of tasks such as UCI datasets and generative models. Unfortunately, the reviewers argued that several important claims remain unsubstantiated. In particular, empirical comparisons with baselines like pseudo-input based methods are missing.